# EDGE-VARYING FOURIER GRAPH NETWORKS FOR MULTIVARIATE TIME SERIES FORECASTING

## ABSTRACT

The key to multivariate time series (MTS) analysis and forecasting is to disclose the underlying couplings between variables that drive the co-movements. Considerable recent successful MTS methods are built with graph neural networks (GNNs) due to their essential capacity for relational modeling. However, previous work often used a static graph structure of time-series variables for modeling MTS, but failed to capture their ever-changing correlations over time. In this paper, we build a fully-connected supra-graph, representing non-static correlations between any two variables at any two timestamps, to capture high-resolution spatial-temporal dependencies. Whereas, conducting graph convolutions on such a supra-graph heavily increases computational complexity. As a result, we propose the novel Edge-Varying Fourier Graph Networks (EV-FGN), which reformulates the graph convolutions in the frequency domain with high efficiency and scale-free parameters, and applies edge varying graph filters to capture the time-varying variable dependencies. Extensive experiments show that EV-FGN outperforms state-of-the-art methods on seven real-world MTS datasets.

## 1 INTRODUCTION

Multivariate time series (MTS) forecasting is a key ingredient in many real-world scenarios, including weather forecasting Zheng et al. (2015), decision making Borovykh et al. (2017), traffic forecasting Yu et al. (2018a); Bai et al. (2020), COVID-19 prediction Cao et al. (2020); Chen et al. (2022), etc. Recently, deep neural networks, such as long short-term memory (LSTM) Hochreiter & Schmidhuber (1997), convolutional neural network (CNN) Borovykh et al. (2017), Transformer Vaswani et al. (2017), have dominated MTS modeling. In particular, graph neural networks (GNNs) have demonstrated promising performance on MTS forecasting with their essential capability to capture the complex couplings between time-series variables. Some studies enable to adaptively learn the graph for MTS forecasting even without an explicit graph structure, e.g., by node similarity Mateos et al. (2019); Bai et al. (2020); Wu et al. (2019) and/or self-attention mechanism Cao et al. (2020).

Despite the success of GNNs on MTS forecasting, three practical challenges are eagerly demanded to address: 1) the dependencies between each pair of time series variables are generally non-static, which demands full dependencies modeling with all possible lags; 2) a high-efficiency dense graph learning method is demanded to replace the high-cost operators of graph convolutions and attention (with quadratic time complexity of the graph size); 3) the graph over MTS varies with different temporal interactions, which demands an efficient dynamic structure encoding method. In this paper, different from most GNN-based methods that construct graphs to model the spatial correlation between variables (i.e. nodes) Bai et al. (2020); Wu et al. (2019; 2020), we attempt to build a supra-graph that sheds light on the "high-resolution" correlations between any two variables at any two timestamps (i.e., fine-grained spatial-temporal dependencies), and largely enhances the expressiveness on non-static spatial-temporal dependencies.

Obviously, the supra-graph will heavily increase the computational complexity of GNN-based model, then a high-efficiency learning method is required to reduce the cost for model training. Inspired by *Fourier Neural Operator* (FNO) Li et al. (2021), we reformulate the graph convolution (time domain) to much lower-complexity matrix multiplication in the frequency domain by leveraging an efficient and newly-defined *Fourier Graph Shift Operator* (FGSO). In addition, it is necessary to consider the multiple iterations (layers) of graph convolutions to expand receptive neighbors and mix the diffusion

information in the graph. To capture time-varying diffusion over the supra-graph, we introduce the edge-varying graph filter to weight the graph edges differently from different iterations and reformulate the edge-varying graph filter with multiple frequency-invariant FGSOs in the frequency domain to reduce the computational cost of graph convolutions. Finally, a novel Edge-Varying Fourier Graph Networks (EV-FGN) is designed for MTS analysis, which is stacked with multiple FGSOs to perform high-efficient multi-layer graph convolutions in the Fourier space.

The main contributions of this paper are summarized as follows:

• We adaptively learn a supra-graph, representing non-static correlations between any two variables at any two timestamps, to capture high-resolution spatial-temporal dependencies.

• To efficiently compute graph convolutions over the supra-graph, we reformulate the graph convolutions in the Fourier space by leveraging FGSO. To the best of our knowledge, this work makes the first step to reformulate the graph convolutions in the Fourier space.

• We design a novel network EV-FGN evolved from edge-varying graph filters for MTS analysis to capture the time-varying variable dependencies in the Fourier space. This study makes the first attempt to design a complex-valued feed-forward network in the Fourier space to efficiently compute multi-layer graph convolutions.

• Extensive experimental results on seven MTS datasets demonstrate that EV-FGN achieves state-of-the-art performance with high efficiency and fewer parameters. Multifaceted visualizations further interpret the efficacy of EV-FGN in graph representation learning for MTS forecasting.

## 2 RELATED WORKS

### 2.1 MULTIVARIATE TIME SERIES FORECASTING

Classic time series forecasting methods are linear models, such as VAR Watson (1993), ARIMA Asteriou & Hall (2011) and state space model (SSM) Hyndman et al. (2008). Recently, deep learning based methods Lai et al. (2018); Sen et al. (2019); Zhou et al. (2021) have dominated MTS forecasting due to their capability of fitting any complex nonlinear correlations Lim & Zohren (2021).

**MTS with GNN**. More recently, MTS have embraced GNN Wu et al. (2019); Bai et al. (2020); Wu et al. (2020); Yu et al. (2018b); Chen et al. (2022); Li et al. (2018) due to their best capability of modeling structural dependencies between variables. Most of these models, such as STGCN Yu et al. (2018b), DCRNN Li et al. (2018) and TAMP-S2GCNets Chen et al. (2022), require a pre-defined graph structure which is usually unknown in most cases. In recent years, some GNN-based works Kipf et al. (2018); Deng & Hooi (2021) account for the dynamic dependencies due to network design such as the time-varying attention Deng & Hooi (2021). In comparison, our proposed model captures the dynamic dependencies leveraging the high-resolution correlation in the supra-graph without introducing specific networks.

**MTS with Fourier transform**. Recently, increasing MTS forecasting models have introduced the Fourier theory into neural networks as high-efficiency convolution operators Guibas et al. (2022); Chi et al. (2020). SFM Zhang et al. (2017) decomposes the hidden state of LSTM into multiple frequencies by discrete Fourier transform (DFT). mWDN Wang et al. (2018) decomposes the time series into multilevel sub-series by discrete wavelet decomposition and feeds them to LSTM network, respectively. ATFN Yang et al. (2022) utilizes a time-domain block to learn the trending feature of complicated non-stationary time series and a frequency-domain block to capture dynamic and complicated periodic patterns of time series data. FEDformer Zhou et al. (2022) proposes an attention mechanism with low-rank approximation in frequency and a mixture of expert decomposition to control the distribution shift. However, these models only capture temporal dependencies in the frequency domain. StemGNN Cao et al. (2020) takes the advantages of both inter-series correlations and temporal dependencies by modeling them in the spectral domain, but, it captures the temporal and spatial dependencies separately. Unlike these efforts, our model is able to jointly encode spatial-temporal dependencies in the Fourier space.

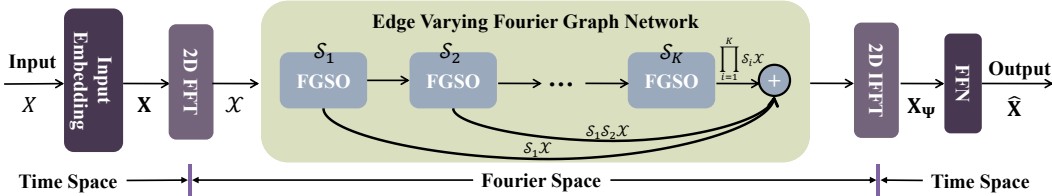

Figure 1: The network architecture of our proposed model. Given an input $X \in \mathbb{R}^{N \times T}$, we 1) embed $X$ into $\mathbf{X} \in \mathbb{R}^{N \times T \times d}$; 2) transform $\mathbf{X}$ to Fourier space $\mathcal{X} \in \mathbb{C}^{N \times T \times d}$ by 2D DFT on the discrete N × T spatial-temporal space; 3) perform graph convolutions in Fourier space by conducting multiplication of FGSO $\mathcal{S}$ and $\mathcal{X}$ in the $K$-layer EV-FGN; 4) transform the output of EV-FGN to time domain $\mathbf{X}_\Psi \in \mathbb{R}^{N \times T \times d}$ by 2D IDFT; 5) generate $\tau$-step predictions $\hat{X} \in \mathbb{R}^{N \times \tau}$ via feeding $\mathbf{X}_\Psi$ to a two-layer feed-forward network.

## 2.2 GRAPH SHIFT OPERATOR

Graph shift operators (GSOs) (e.g., the adjacency matrix and the Laplacian matrix) are a general set of linear operators which are used to encode neighbourhood topologies in the graph. Klicpera et al. (2019) shows that applying the varying GSOs in the message passing step of GNNs can lead to significant improvement of performance. Dasoulas et al. (2021) proposes a parameterized graph shift operator to automatically adapt to networks with varying sparsity. Isufi et al. (2021) allows different nodes to use different GSOs to weight the information of different neighbors. Hadou et al. (2022) introduces a linear composition of the graph shift operator and time-shift operator to design space-time filters for time-varying graph signals. Inspired by these works, in this paper we design a varying parameterized graph shift operator in Fourier space.

## 2.3 FOURIER NEURAL OPERATOR

Different from classical neural networks which learn mappings between finite-dimensional Euclidean spaces, neural operators learn mappings between infinite-dimensional function spaces Kovachki et al. (2021b). Fourier neural operators (FNOs), currently the most promising one of the neural operators, are universal, in the sense that they can approximate any continuous operator to the desired accuracy Kovachki et al. (2021a). Li et al. (2021) formulates a new neural operator by parameterizing the integral kernel directly in the Fourier space, allowing for an expressive and efficient architecture for partial differential equations. Guibas et al. (2022) proposes an efficient token mixer that learns to mix in the Fourier domain which is a principled architectural modification to FNO. In this paper, we learn a Fourier graph shift operator by leveraging the Fourier Neural operator.

## 3 METHODOLOGY

Let us denote the entire MTS raw data as $\mathbb{X} \in \mathbb{R}^{N \times L}$ with $N$ variables and $L$ timestamps. Under the rolling setting, we have window-sized time-series inputs with $T$ timestamps, i.e., $\{X | X \subset \mathbb{X}, X \in \mathbb{R}^{N \times T}\}$. Accordingly, we formulate the problem of MTS forecasting as learning the spatial-temporal dependencies simultaneously on a supra-graph $\mathcal{G} = (X, S)$ attributed to each $X$. The supra-graph $\mathcal{G}$ contains $N * T$ nodes that represent values of each variable at each timestamp in $X$, and $S \in \mathbb{R}^{(N*T) \times (N*T)}$ is a graph shift operator (GSO) representing the connection structure of $\mathcal{G}$. Since the underlying graph is unknown in most MTS scenarios, we assume all nodes in the supra-graph are connected with each other, i.e., a fully-connected graph, and perform graph convolutions on the graph to learn spatial-temporal representation. Then, given the observed values of previous $T$ steps at timestamp $t$, i.e., $X^{t-T:t} \in \mathbb{R}^{N \times T}$, the task of *multi-step multivariate time series forecasting* is to predict the values of $N$ variables for next $\tau$ steps denoted as $\hat{X}^{t+1:t+\tau} \in \mathbb{R}^{N \times \tau}$ on the supra-graph $\mathcal{G}$, formulated as follows:

$$\hat{X}^{t+1:t+\tau} = \mathrm{F}(X^{t-T:t}; \mathcal{G}; \Theta) \qquad (1)$$

where F is the forecasting model with parameters $\Theta$.

### 3.1 OVERALL ARCHITECTURE

The overall architecture of our model is illustrated in Fig. 1. Given input data $X \in \mathbb{R}^{N \times T}$, first we embed the data into embeddings $\mathbf{X} \in \mathbb{R}^{N \times T \times d}$ by assigning a $d$-dimension vector for each node using an embedding matrix $\Phi \in \mathbb{R}^{N \times T \times d}$, i.e., $\mathbf{X} = X \times \Phi$. Instead of directly learning the huge embedding matrix, we introduce two small parameter matrices: 1) a variable embedding matrix $\phi_v \in \mathbb{R}^{N \times 1 \times d}$, and 2) a temporal embedding matrix $\phi_u \in \mathbb{R}^{1 \times T \times d}$ to factorize $\Phi$, i.e., $\Phi = \phi_v \times \phi_u$. Subsequently, we perform 2D discrete Fourier transform (DFT) on each discrete $N \times T$ spatial-temporal plane of the embeddings $\mathbf{X}$ and obtain the frequency input $\mathcal{X} := \mathrm{DFT}(\mathbf{X}) \in \mathbb{C}^{N \times T \times d}$. We then feed $\mathcal{X}$ to $K$-layer Edge-Varying Fourier Graph Networks (denoted as $\Psi_K$) to perform graph convolutions for capturing the spatial-temporal dependencies simultaneously in the Fourier space.

To make predictions in time domain, we perform 2D inverse Fourier transform to generate representations $\mathbf{X}_\Psi := \mathrm{IDFT}(\Psi_K(\mathcal{X})) \in \mathbb{R}^{N \times T \times d}$ in the time domain. The representation is then fed to two-layer feed-forward networks (FFN, see more details in Appendix F.4) parameterized with weights and biases denoted as $\phi_{ff}$ to make predictions for future $\tau$ steps $\hat{X} \in \mathbb{R}^{N \times \tau}$ by one forward procedure. The L2 loss function for multi-step forecasting can be formulated as:

$$\mathcal{L}(\hat{X}; X; \Theta) = \sum_t \left\| \hat{X}^{t+1:t+\tau} - X^{t+1:t+\tau} \right\|_2^2 \tag{2}$$

with parameters $\Theta = \{\phi_v, \phi_u, \phi_{ff}, \Psi_K\}$ and the groudtruth $X^{t+1:t+\tau} \subset \mathbb{X}$ at timestamp $t$.

### 3.2 FOURIER GRAPH SHIFT OPERATOR

According to the discrete signal processing on graphs Sandryhaila & Moura (2013), a graph shift operator (GSO) is defined as a general family of operators which enables the diffusion of information over graph structures Gama et al. (2020); Dasoulas et al. (2021).

**Definition 1 (Graph Shift Operator).** *Given a graph $G$ with $n$ nodes, a matrix $S \in \mathbb{R}^{n \times n}$ is called a Graph Shift Operator (GSO) if it satisfies $S_{ij} = 0$ if $i \neq j$ and nodes $i, j$ are not connected.*

The graph shift operator includes the adjacency, Laplacian matrices and their normalisations as instances of its class, and represents the connection structure of the graph. Accordingly, given the graph $G$ attributed to $X \in \mathbb{R}^{n \times d}$ (corresponding to the supra-graph with $n = N * T$), a general form of spatial-based graph convolution is defined as

$$O(X) := SXW \tag{3}$$

with the parameter matrix $W \in \mathbb{R}^{d \times d}$. Regarding $S$ as $n \times n$ scores, we can define a matrix-valued kernel $\kappa : [n] \times [n] \to \mathbb{R}^{d \times d}$ with $\kappa[i,j] = S_{ij} \circ W$, where $[n] = \{1, 2, \cdots, n\}$. Then the graph convolution can be viewed as a kernel summation.

$$O(X)[i] = \sum_{j=1}^n X[j]\kappa[i,j] \qquad \forall i \in [n]. \tag{4}$$

In the special case of the Green's kernel $\kappa[i,j] = \kappa[i-j]$, we can rewrite the kernel summation

$$O(X)[i] = \sum_{j=1}^n X[j]\kappa[i-j] = (X * \kappa)[i] \qquad \forall i \in [n]. \tag{5}$$

with $X * \kappa$ denotes the convolution of discrete sequences $X$ and $\kappa$. According to the convolution theorem Katznelson (1970) (see Appendix B), the graph convolution is rewritten as

$$O(X)(i) = \mathcal{F}^{-1}\left(\mathcal{F}(X)\mathcal{F}(\kappa)\right)(i) \qquad \forall i \in [n]. \tag{6}$$

where $\mathcal{F}$ and $\mathcal{F}^{-1}$ denote the discrete Fourier transform (DFT) and its inverse (IDFT), respectively. The multiplication in the Fourier space is a lower-complexity computation compared to the graph convolution, and DFT can be efficiently implemented by the fast Fourier transform (FFT).

**Definition 2 (Fourier Graph Shift Operator).** *Given a graph $G = (X, S)$ with input $X \in \mathbb{R}^{n \times d}$ and GSO $S \in \mathbb{R}^{n \times n}$ and the weight matrix $W \in \mathbb{R}^{d \times d}$, the graph convolution is formulated as*

$$\mathcal{F}(SXW) = \mathcal{F}(X) \times_n \mathcal{F}(\kappa) \tag{7}$$

*where $\mathcal{F}$ denotes DFT, satisfies $\kappa[i,j] = \kappa[i-j]$, and $\times_n$ is matrix multiplication on dimensions except that of $n$. We define $\mathcal{S} := \mathcal{F}(\kappa) \in \mathbb{C}^{n \times d \times d}$ as a Fourier graph shift operator (FGSO).*

In particular, turning to our case of the fully-connected supra-graph $\mathcal{G}$ with an all-one GSO $S \in \{1\}^{n \times n}$, it yields the space-invariant kernel $\kappa[i, j] = S_{ij} \circ W = W$ and $\mathcal{F}(SXW) = \mathcal{F}(X)\mathcal{F}(\kappa)$. Accordingly, we can parameterize FGSO $\mathcal{S}$ with a complex-valued matrix $\mathbb{C}^{d \times d}$ which is frequency-invariant and is computationally low costly compared to a varying kernel resulting a parameterized matrix of $\mathbb{C}^{n \times d \times d}$. Furthermore, we can extend Definition 2 to 2D discrete space, i.e., from $[n]$ to $[N] \times [T]$, corresponding to the finite discrete spatial-temporal space of multivariate time series. See Appendix C for more explanations on the frequency-invariant FGSO and the extension to 2D domain.

**Remarks**. Compared with FGSO and FNO, frequency-invariant FGSO has several advantages. Assume a graph with $n$ nodes and the embedding dimension $d$ ($d \le n$). 1) *Efficiency*: the time complexity of frequency-invariant FGSO is $O(nd \log n + nd^2)$ for DFT, IDFT and the matrix multiplication compared with that of $O(n^2d + nd^2)$ on a GSO. 2) *Scale-free parameters*: frequency-invariant FGSO strategically shares $O(d^2)$ parameters for each node and the parameter volume is agnostic to the data scale, while the parameter count of FNO is $O(nd^2)$.

### 3.3 EDGE-VARYING FOURIER GRAPH NETWORKS

Graph filters as core operations in signal processing are linear transformations expressed as polynomials of the graph shift operator Isufi et al. (2021); Mateos et al. (2019) and can be used to exactly model graph convolutions and capture multi-order diffusion on graph structures Segarra et al. (2017). To capture the time-varying counterparts and adopt different weights to weight the information of different neighbors in each diffusion order, the edge-variant graph filters are defined as follows Isufi et al. (2021); Segarra et al. (2017): given GSO $S \in \mathbb{R}^{n \times n}$ corresponding to a graph with $n$ nodes,

$$H_{EV} = S_0 + S_1 S_0 + ... + S_K S_{K-1}...S_0 = \sum_{k=0}^{K} S_{k:0} \tag{8}$$

where $S_0$ denotes the identity matrix, $\{S_k\}_{k=1}^{K}$ is a collection of $K$ edge-weighting GSOs sharing the sparsity pattern of $S$, and $S_k \in \mathbb{R}^{n \times n}$ corresponds to the $k$-th diffusion step. The edge-variant graph filters are proved effective to yield a highly discriminative model and lay the foundation for the unification of GCNs and GATs Isufi et al. (2021). We can extend Equation 7 to $H_{EV}$ and reformulate the multi-order graph convolution in a recursive form, where we omit the weight $W$ for conciseness.

**Proposition 1.** *Given a graph input $X \in \mathbb{R}^{n \times d}$, the $K$-order graph convolution under the edge-variant graph filter $H_{EV} = \sum_{k=0}^{K} S_{k:0}$ with $\{S_k \in \mathbb{R}^{n \times n}\}_{k=0}^{K}$ is reformulated as follows:*

$$H_{EV}X = \mathcal{F}^{-1}\left(\sum_{k=0}^{K} \mathcal{F}(X) \times_n \mathcal{S}_{0:k}\right) \quad s.t. \ \mathcal{S}_{0:k} = \mathcal{S}_0 \times_n \cdots \mathcal{S}_{K-1} \times_n \mathcal{S}_K \tag{9}$$

*where $\mathcal{S}_k \in \mathbb{C}^{n \times d \times d}$ is the $k$-th FGSO satisfying $\mathcal{F}(S_k X) = \mathcal{F}(X) \times_n \mathcal{S}_k$, $\mathcal{S}_0$ is the identity matrix, and $\mathcal{F}$ and $\mathcal{F}^{-1}$ denote the discrete Fourier transform and its reverse, respectively.*

Proposition 1 proved in Appendix D.1 states that we can rewrite the multi-order graph convolution corresponding to $H_{EV}$ as a summation of a recursive multiplication of individual FGSO in the Fourier space. Corresponding to our case of the supra-graph $\mathcal{G}$, we can similarly adopt $K$ frequency-invariant FGSOs and parameterize each FGSO $\mathcal{S}_k$ with a complex-valued matrix $\mathbb{C}^{d \times d}$. This saves a large amount of computation costs and results in a concise form:

$$H_{EV}X = \mathcal{F}^{-1}\left(\sum_{k=0}^{K} \mathcal{F}(X)\mathcal{S}_{0:k}\right) \quad s.t. \ \mathcal{S}_{0:k} = \prod_{i=0}^{k} \mathcal{S}_i \tag{10}$$

The recursive composition has a nice property that $\mathcal{S}_{0:k} = \mathcal{S}_{0:k-1}\mathcal{S}_k$, which inspires us to design a complex-valued feed forward network with $\mathcal{S}_k$ being the complex weights for the $k$-th layer. However, both the FGSO and edge-variant graph filter are linear transformations, limiting the capability of modeling non-linear information diffusion on graphs. Following by the convention in GNNs, we introduce the non-linear activation and biases to reformulate the $k$-the layer as follows:

$$\mathcal{X}_k = \sigma(\mathcal{X}_{k-1}\mathcal{S}_k + b_k) \tag{11}$$

with the complex weight $\mathcal{S}_k \in \mathbb{C}^{d \times d}$, biases $b_k \in \mathbb{C}^d$ and the activation function $\sigma$. Accordingly, we design an edge-varying Fourier graph network (EV-FGN) in the Fourier space according to Equations

9 and 11, as shown in Fig. 1. Accordingly, the $K$-layer EV-FGN $\Psi_K$ is formulated:

$$\Psi_K(\mathcal{X}) = \sum_{k=0}^{K} \mathcal{X}_k \quad s.t. \ \mathcal{X}_k = \sigma(\mathcal{X}_{k-1}\mathcal{S}_k + b_k) \tag{12}$$

where $\mathcal{X}_0 := \mathcal{F}(\mathbf{X})$, $\{\mathcal{S}_k \in \mathbb{C}^{d \times d}\}_{k=1}^{K}$ and $\{b_k \in \mathbb{C}^d\}_{k=1}^{K}$ are complex-valued parameters. The frequency output $\Psi_K(\mathcal{X})$ is then transformed via IDFT, followed with feed forward networks (see Appendix F.4 for more details) to make multi-step forecasting in the time domain.

**Remarks**. EV-FGN efficiently learns edge- and neighbor-dependent weights to compute multi-order graph convolutions on one fly in the Fourier space with scale-free parameter volume. For $K$ iterations of graph convolutions, GCNs have a general time complexity of $O(Kn^2d + Knd^2)$, and FNO needs $O(Knd \log n + Knd^2)$, and EV-FGN achieves a time complexity of $O(nd \log n + Knd^2 + Knd)$. EV-FGN saves time costs from FNO in Fourier transforms and is much more efficient than GCNs, especially in our case with $n = NT$ for a given raw input $X \in \mathbb{R}^{N \times T}$. See Appendix E for more details about the relations and differences between EV-FGN with FNO, adaptive FNO, and GNNs.

## 4 EXPERIMENTS

### 4.1 SETUP

**Datasets**. We select seven representative datasets from different application scenarios for evaluation, including traffic, energy, web traffic, electrocardiogram, and COVID-19. These datasets are summarized in Table 1. All datasets are normalized using the min-max normalization. Except the COVID-19 dataset, we split the other datasets into training, validation, and test sets with the ratio of 7:2:1 in chronological order. For the COVID-19 dataset, the ratio is 6:2:2.

Table 1: Summary of datasets.

| Datasets | Solar | Wiki | Traffic | ECG | Electricity | COVID-19 | METR-LA |
|---|---|---|---|---|---|---|---|
| Samples | 3650 | 803 | 10560 | 5000 | 140211 | 335 | 34272 |
| Variables | 592 | 2000 | 963 | 140 | 370 | 55 | 207 |
| Granularity | 1hour | 1day | 1hour | - | 15min | 1day | 5min |
| Start time | 01/01/2006 | 01/07/2015 | 01/01/2015 | - | 01/01/2011 | 01/02/2020 | 01/03/2012 |

**Baselines**. We compare the forecasting performance of our EV-FGN with other representative and SOTA models on the seven datasets, including VAR Watson (1993), SFM Zhang et al. (2017), LSTNet Lai et al. (2018), TCN Bai et al. (2018), GraphWaveNet Wu et al. (2019), DeepGLO Sen et al. (2019), StemGNN Cao et al. (2020), MTGNN Wu et al. (2020), AGCRN Bai et al. (2020), Reformer Kitaev et al. (2020), Informer Zhou et al. (2021), Autoformer Wu et al. (2021), FEDformer Zhou et al. (2022), and CoST Woo et al. (2022). In addition, we compare EV-FGN with SOTA TAMP-S2GCNets Chen et al. (2022), DCRNN Li et al. (2018) and STGCN Yu et al. (2018a), which need pre-defined graph structures.

**Experimental Settings**. All experiments are implemented in Python in Pytorch 1.8 (SFM in Keras) and conducted on one NVIDIA RTX 3080 card. Our model is trained using RMSProp with a learning rate of 0.00001 and MSELoss (Mean Squared Error) as the loss function. The best parameters for all comparative models are chosen through careful parameter tuning on the validation set. We use MAE (Mean Absolute Errors), RMSE (Root Mean Squared Errors), and MAPE (Mean Absolute Percentage Error) to measure the performance.

More details about the datasets, baselines and experimental settings can be found in Appendix F.

### 4.2 RESULTS

We summarize the evaluation results in Table 2, and more results can be found in Appendix G. Generally, our model EV-FGN establishes a new state-of-the-art on all datasets. On average, EV-FGN improves $8.8\%$ on MAE and $10.5\%$ on RMSE compared to the best baseline for all datasets. Among these baselines, Reformer, Informer, Autoformer and FEDformer are transformer-based models

that achieve competitive performance on Electricity and COVID-19 datasets since those models are competent in capturing temporal dependencies. However, they are defective in capturing the spatial dependencies explicitly. GraphWaveNet, MTGNN, StemGNN and AGCRN are GNN-based models that show promising performances on Wiki, Traffic, Solar and ECG datasets due to their high capability in handling spatial dependencies among variables. However, they are limited to simultaneously capturing spatial-temporal dependencies. EV-FGN significantly outperforms the baseline models since it learns comprehensive spatial-temporal dependencies simultaneously and attends to time-varying dependencies among variables. In Appendix G, we further report the results on four datasets and show the comparison between EV-FGN with those models requiring pre-defined graph structures.

Table 2: Forecasting results on the six datasets.

| Models | Solar | | | Wiki | | | Traffic | | |
|---|---|---|---|---|---|---|---|---|---|
| | MAE | RMSE | MAPE(%) | MAE | RMSE | MAPE(%) | MAE | RMSE | MAPE(%) |
| VAR Watson (1993) | 0.184 | 0.234 | 577.10 | 0.057 | 0.094 | 96.58 | 0.535 | 1.133 | 550.12 |
| SFM Zhang et al. (2017) | 0.161 | 0.283 | 362.89 | 0.081 | 0.156 | 104.47 | 0.029 | 0.044 | 59.33 |
| LSTNet Lai et al. (2018) | 0.148 | 0.200 | 132.95 | 0.054 | 0.090 | 118.24 | 0.026 | 0.057 | **25.77** |
| TCN Bai et al. (2018) | 0.176 | 0.222 | 142.23 | 0.094 | 0.142 | 99.66 | 0.052 | 0.067 | - |
| DeepGLO Sen et al. (2019) | 0.178 | 0.400 | 346.78 | 0.110 | 0.113 | 119.60 | 0.025 | 0.037 | 33.32 |
| Reformer Kitaev et al. (2020) | 0.234 | 0.292 | 128.58 | 0.048 | 0.085 | 73.61 | 0.029 | 0.042 | 112.58 |
| Informer Zhou et al. (2021) | 0.151 | 0.199 | 128.45 | 0.051 | 0.086 | 80.50 | 0.020 | 0.033 | 59.34 |
| Autoformer Wu et al. (2021) | 0.150 | 0.193 | 103.79 | 0.069 | 0.103 | 121.90 | 0.029 | 0.043 | 100.02 |
| FEDformer Zhou et al. (2022) | 0.139 | 0.182 | **100.92** | 0.068 | 0.098 | 123.10 | 0.025 | 0.038 | 85.12 |
| GraphWaveNet Wu et al. (2019) | 0.183 | 0.238 | 603 | 0.061 | 0.105 | 136.12 | 0.013 | 0.034 | 33.78 |
| StemGNN Cao et al. (2020) | 0.176 | 0.222 | 128.39 | 0.190 | 0.255 | 117.92 | 0.080 | 0.135 | 64.51 |
| MTGNN Wu et al. (2020) | 0.151 | 0.207 | 507.91 | 0.101 | 0.140 | 122.96 | 0.013 | 0.030 | 29.53 |
| AGCRN Bai et al. (2020) | 0.123 | 0.214 | 353.03 | 0.044 | 0.079 | 78.52 | 0.084 | 0.166 | 31.73 |
| **EV-FGN(ours)** | **0.120** | **0.162** | 116.48 | **0.041** | **0.076** | **64.50** | **0.011** | **0.023** | 28.71 |
| Improvement | 2.4% | 11.0% | - | 6.8% | 3.8% | 12.4% | 15.4% | 23.3% | - |
| Models | ECG | | | Electricity | | | COVID-19 | | |
| | MAE | RMSE | MAPE(%) | MAE | RMSE | MAPE(%) | MAE | RMSE | MAPE(%) |
| VAR Watson (1993) | 0.120 | 0.170 | 22.56 | 0.101 | 0.163 | 43.11 | 0.226 | 0.326 | 191.95 |
| SFM Zhang et al. (2017) | 0.095 | 0.135 | 24.20 | 0.086 | 0.129 | 33.71 | 0.205 | 0.308 | 76.08 |
| LSTNet Lai et al. (2018) | 0.079 | 0.115 | 18.68 | 0.075 | 0.138 | 29.95 | 0.248 | 0.305 | 89.04 |
| TCN Bai et al. (2018) | 0.078 | 0.107 | 17.59 | 0.057 | 0.083 | 26.64 | 0.317 | 0.354 | 151.78 |
| DeepGLO Sen et al. (2019) | 0.110 | 0.163 | 43.90 | 0.090 | 0.131 | 29.40 | 0.169 | 0.253 | 75.19 |
| Reformer Kitaev et al. (2020) | 0.062 | 0.090 | 13.58 | 0.078 | 0.129 | 33.37 | 0.152 | 0.209 | 132.78 |
| Informer Zhou et al. (2021) | 0.056 | 0.085 | 11.99 | 0.070 | 0.119 | 32.66 | 0.200 | 0.259 | 155.55 |
| Autoformer Wu et al. (2021) | 0.055 | 0.081 | 11.37 | 0.056 | 0.083 | 25.94 | 0.159 | 0.211 | 136.24 |
| FEDformer Zhou et al. (2022) | 0.055 | 0.080 | 11.16 | 0.055 | 0.081 | 25.84 | 0.160 | 0.219 | 134.45 |
| GraphWaveNet Wu et al. (2019) | 0.093 | 0.142 | 40.19 | 0.094 | 0.140 | 37.01 | 0.201 | 0.255 | 100.83 |
| StemGNN Cao et al. (2020) | 0.100 | 0.130 | 29.62 | 0.070 | 0.101 | - | 0.421 | 0.508 | 141.01 |
| MTGNN Wu et al. (2020) | 0.090 | 0.139 | 35.04 | 0.077 | 0.113 | 29.77 | 0.394 | 0.488 | 88.13 |
| AGCRN Bai et al. (2020) | 0.055 | 0.080 | 11.75 | 0.074 | 0.116 | 26.08 | 0.254 | 0.309 | **58.58** |
| **EV-FGN(ours)** | **0.052** | **0.078** | **11.05** | **0.051** | **0.077** | **24.28** | **0.129** | **0.173** | 71.52 |
| Improvement | 5.5% | 2.5% | 1.0% | 7.3% | 4.9% | 6.0% | 15.1% | 17.2% | - |

## 4.3 ANALYSIS

**Efficiency Analysis**. We investigate the parameter volumes and training time costs of EV-FGN, StemGNN, AGCRN, GraphWaveNet and MTGNN on two representative datasets (Wiki and Traffic). We report the parameter volumes and the average time costs of five rounds of experiments in Table 3. In terms of parameter volumes, EV-FGN requires the least volume of parameters among the comparative models. Specifically, it achieves 32.2% and 9.5% parameter reduction over GraphWaveNet on Traffic and Wiki datasets, respectively. This is highly attributed that EV-FGN has shared scale-free parameters for each node. Regarding the training time, EV-FGN runs much faster than all baseline models, and it shows 5.8% and 23.3% efficiency improvements over the fast baseline GraphWaveNet on Traffic and Wiki datasets, respectively. Besides, the variable count of Wiki dataset is about twice larger than that of Traffic dataset, but EV-FGN shows larger efficiency gaps with the baselines. These results demonstrate that EV-FGN shows high efficiency in computing graph convolutions and is scalable to large datasets with large graphs. **Significantly**, the supra-graph in EV-FGN has $N \times T$ nodes, which is much larger than the graphs (with $N$ nodes) in the baselines.

**Ablation study**. We conduct the ablation study on METR-LA dataset to evaluate the contribution of different components of our model. The results shown in Table 4 verify the effectiveness of

Table 3: Results of parameter volumes and training time costs on Traffic and Wiki datasets.

| models | Traffic | | Wiki | |
| --- | --- | --- | --- | --- |
| | Parameters | Training (s/epoch) | Parameters | Training (s/epoch) |
| StemGNN | 1606140 | 185.86±2.22 | 4102406 | 92.95±1.39 |
| MTGNN | 707516 | 169.34±1.56 | 1533436 | 28.69±0.83 |
| AGCRN | 749940 | 113.46±1.91 | 755740 | 22.48±1.01 |
| GraphWaveNet | 280860 | 105.38±1.24 | 292460 | 21.23±0.76 |
| EV-FGN (ours) | 190564 | 99.25±1.07 | 264804 | 16.28±0.48 |

each component. Specifically, **w/o Embedding** shows the significance of performing embedding to improve model generalization. **w/o Dynamic Filter** using the same graph shift operator verifies the effectiveness of applying different graph shift operators in capturing time-varying dependencies. In addition, **w/o Residual** represents EV-FGN without the $K = 0$ layer, while **w/o Summation** adopts the last order (layer) output $\mathcal{X}_K$ as the output of EV-FGN. These results demonstrate the importance of high-order diffusion and the contribution of multi-order diffusion. More results and analysis of the ablation study are provided in Appendix H.1.

Table 4: Ablation study on the METR-LA dataset.

| metrics | w/o Embedding | w/o Dynamic Filter | w/o Residual | w/o Summation | EV-FGN |
| --- | --- | --- | --- | --- | --- |
| MAE | 0.053 | 0.055 | 0.054 | 0.054 | 0.050 |
| RMSE | 0.116 | 0.114 | 0.115 | 0.114 | 0.113 |
| MAPE(%) | 86.73 | 86.69 | 86.75 | 86.62 | 86.30 |

## 4.4 VISUALIZATION

To better investigate the spatial-temporal representation learned by EV-FGN, we conduct more visualization experiments on METR-LA and COVID-19 datasets. More details about the way of visualization can be found in Appendix F.5.

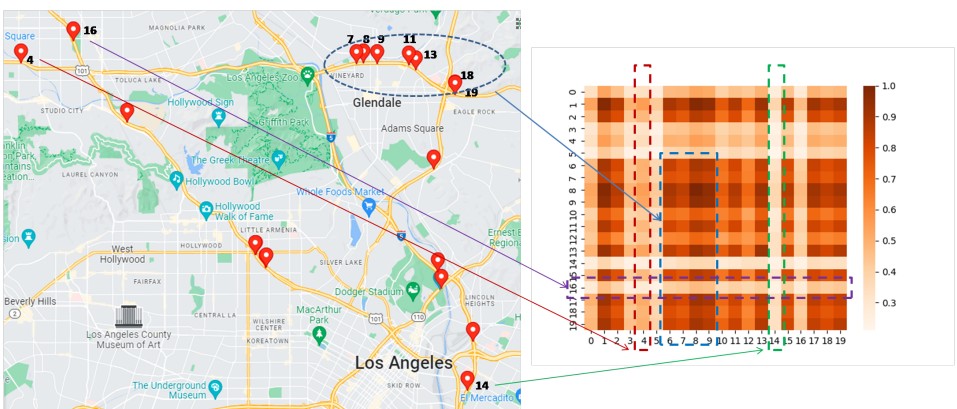

Figure 2: The adjacency matrix (right) learned by EV-FGN and the corresponding road map (left).

**Visualization of spatial representations learned by EV-FGN**. We produce the visualization of the generated adjacency matrix according to the learned representation from EV-FGN on the METR-LA dataset. Specifically, we randomly select 20 detectors and visualize their corresponding adjacency matrix via heat map, as shown in Fig. 2. Correlating the adjacency matrix with the real road map, we observe: 1) the detectors (7, 8, 9, 11, 13, 18, and 19) are very close in physical distance, corresponding to the high values of their correlations with each other in the heat map; 2) the detectors 4, 14 and 16 have small overall correlation values since they are far from other detectors; 3) **however**, compared with detectors 14 and 16, the detector 4 has slightly higher correlation values to other detectors e.g.,

7, 8, 9, which is attributed that although they are far apart, the detectors 4, 7, 8, 9 are on the same road. The results verify the effectiveness of EV-FGN in learning highly interpretative correlations.

**Visualization on four consecutive days**. Furthermore, we conduct experiments to visualize the adjacency matrix of 10 randomly-selected counties on four consecutive days on the COVID-19 dataset. The visualization results via heat map are shown in Fig. 3. From the figure, we can observe that EV-FGN learns clear spatial patterns that show continuous evolution in the time dimension. This is because EV-FGN highlights the edge-varying design and attends to the time-varying variability of the supra-graph. These results verify that our model enjoys the feasibility of exploiting the time-varying dependencies among variables.

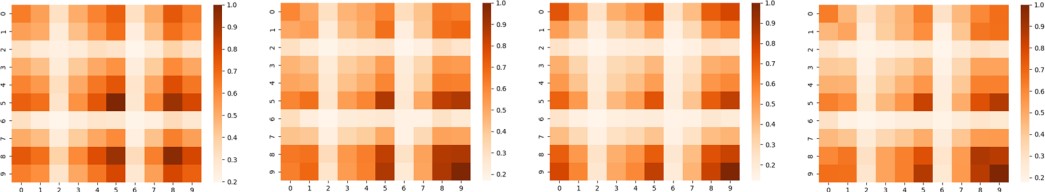

Figure 3: The adjacency matrix for four consecutive days on the COVID-19 dataset.

**Visualization of EV-FGN diffusion process**. To understand how EV-FGN works, we analyze the frequency input of each layer. We choose 10 counties from COVID-19 dataset and visualize their adjacency matrices at two different timestamps, as shown in Fig. 4. From left to right, the results correspond to $\mathcal{X}_0, \cdots, \mathcal{X}_3$ respectively. From the top, we can find that as the number of layers increases, some correlation values are reduced, indicating that some correlations are filtered out. In contrast, the bottom case illustrates some correlations are enhanced as the number of layers increases. These results show that EV-FGN can adaptively and effectively capture important patterns while removing noises to a learn discriminative model. More visualizations are provided in Appendix H.2.

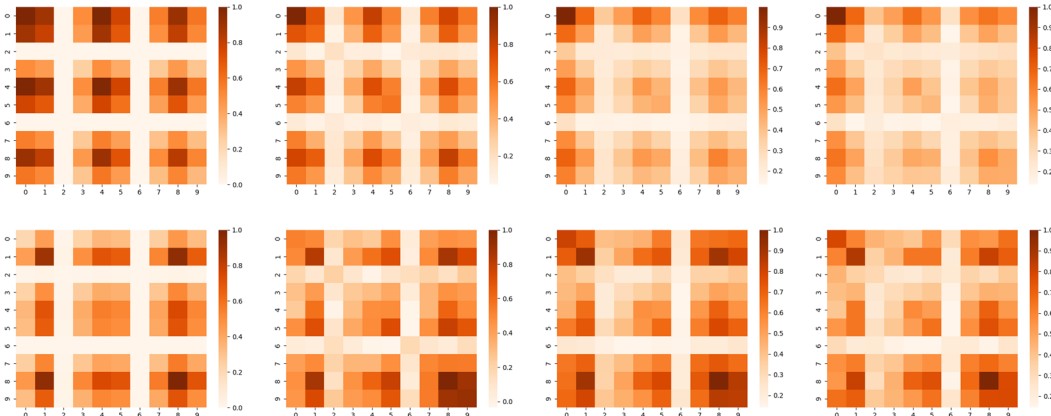

Figure 4: The diffusion process of EV-FGN at two timestamps (top and bottom) on COVID-19.

## 5 CONCLUSION

In this paper, we define a Fourier graph shift operator (FGSO) and construct the efficient edge-varying Fourier graph networks (EV-FGN) for MTS forecasting. EV-FGN is adopted to simultaneously capture high-resolution spatial-temporal dependencies and account for time-varying variable dependencies. This study makes the first attempt to design a complex-valued feed-forward network in the Fourier space for efficiently computing multi-layer graph convolutions. Extensive experiments demonstrate that EV-FGN achieves state-of-the-art performances with higher efficiency and fewer parameters and shows high interpretability in graph representation learning. This study sheds light on efficiently calculating graph operations in Fourier space by learning a Fourier graph shift operator.

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

## A  NOTATION

Table 5: Notations

| | |
|---|---|
| $\mathbb{X}$ | entire multivariate time series data, $\mathbb{X} \in \mathbb{R}^{N \times L}$ |
| $X$ | multivariate time series input under the rolling setting, $X \in \mathbb{R}^{N \times T}$ |
| $N$ | the number of variables of $X$ |
| $L$ | the number of timestamps of $\mathbb{X}$ |
| $T$ | the number of timestamps of X |
| $\tau$ | the prediction length of MTS under the rolling setting |
| $d$ | the embedding dimension |
| $\mathbf{X}$ | the embedding of X, $\mathbf{X} \in \mathbb{R}^{N \times T \times d}$ |
| $\mathcal{X}$ | the spectrum of $X$, $\mathcal{X} \in \mathbb{C}^{N \times T \times d}$ |
| $S$ | the graph shift operator |
| $\mathcal{G}$ | the supra-graph, $\mathcal{G} = \{X, S\}$ attributed to $X$ |
| $\mathcal{S}$ | the Fourier graph shift operator |
| $\Theta$ | the parameters of the forecasting model |
| $\Phi$ | the embedding matrix |
| $\phi_v$ | the variable embedding matrix |
| $\phi_u$ | the temporal embedding matrix |
| $K$ | the diffusion steps |
| $\kappa$ | the kernel function |
| $W$ | the weight matrix |
| $b$ | the complex bias weights |
| $\mathcal{F}$ | Discrete Fourier transform |
| $\mathcal{F}^{-1}$ | Inverse discrete Fourier transform |
| F | the forecasting model |
| $\Psi_K$ | $K$-layer EV-FGN |
| $H_{EV}$ | the edge-varying graph filters |

## B  CONVOLUTION THEOREM

The convolution theorem Katznelson (1970) is one of the most important property of Fourier transform. It states the Fourier transform of a convolution of two signals equals the pointwise product of their Fourier transforms in the frequency domain. Given a signal $x[n]$ and a filter $h[n]$, the convolution theorem can be defined as follows:

$$\mathcal{F}((x * h)[n]) = \mathcal{F}(x)\mathcal{F}(h) \tag{13}$$

where $(x * h)[n] = \sum_{m=0}^{N-1} h[m]x[(n - m)_N]$ denotes the convolution of $x$ and $h$, $(n - m)_N$ denotes $(n - m)$ modulo N, and $\mathcal{F}(x)$ and $\mathcal{F}(h)$ denote discrete Fourier transform of $x[n]$ and $h[n]$, respectively.

## C  EXPLANATIONS

### C.1  THE INTERPRETATION OF FREQUENCY-INVARIANT FGSO

In addition to reducing parameter volumes and saving computation costs, the frequency-invariant parameterized FGSO is empirically proved effective to improve model generalization. As we mentioned above, we perform EV-FGN over the input embeddings and adopt the frequency-invariant FGSOs. Relatively to directly adopting the frequency-variant FGSOs ($\mathbb{C}^{n \times d \times d}$), we subtly "factorize" the frequency-variant parameterization to the time domain (i.e., the embedding matrix $\Phi \in \mathbb{R}^{n \times d}$) and the frequency domain (i.e., frequency-invariant FGSO $\mathbb{C}^{d \times d}$). In the time domain, we embed the raw MTS inputs to improve the model learning capability, while we learn the same transformation (FGSO) for all $N * T$ frequency points in the frequency domain (similar to CNN with shared-weight

convolution kernels or filters that slide along input features). Note that the frequency spectrum in the frequency domain has a global view of which each frequency point attends to all variables or timestamps. This treatment in EV-FGN guarantees the model capacity and is empirically proved superior over the treatment without embeddings and with frequency-variant FGSO (please refer to the Ablation study for detailed results). Although the frequency-variant parameterization may be more powerful and flexible than the frequency-invariant one, it introduces more parameters in the frequency domain, especially for multi-layer EV-FGN, and may not obtain superior performance due to inadequate training or overfitting.

### C.2 EXPLANATION TO THE EXTENSION OF DEFINITION 2 TO 2D DOMAIN

Recall Eqs. 4 and 5,

$$\text{Eq. 4 (kernel summation): } O(X)[i] = \sum_{j=1}^{n} X[j]\kappa[i,j] \qquad \forall i \in [n]$$
$$\text{Eq. 5 (kernel summation): } O(X)[i] = \sum_{j=1}^{n} X[j]\kappa[i-j] = (X * \kappa)[i] \qquad \forall i \in [n]$$

When we extend the equations to 2D domain, i.e., from $[n]$ to $[N] \times [T]$, it means performing a kernel summation/graph convolution over the discrete spatial-temporal space corresponding to all nodes in the supra-graph. Obviously, these computations of the kernel summation can be easily extended to 2D domain. Similarly, according to the convolution theorem, we can obtain a 2D-version of Eq. 6:

$$\text{Eq. 6 (graph convolution): } O(X)(i) = \mathcal{F}^{-1}\left(\mathcal{F}(X)\mathcal{F}(\kappa)\right)(i) \qquad \forall i \in [N] \times [T]$$

with $\mathcal{F}$ and $\mathcal{F}^{-1}$ denote the 2D discrete Fourier transform and its inverse, respectively. Accordingly, when extending Definition 2 to 2D domain, $\mathcal{F}$ denotes the 2D discrete Fourier transform. Therefore, given input embeddings $\mathbf{X} \in \mathbb{R}^{N \times T \times d}$, we perform 2D discrete Fourier transform on each discrete $N \times T$ spatial-temporal plane of the embeddings to obtain the frequency spectrum, and then feed the frequency input into K-layer EV-FGN followed by two-layer (real-valued) feed-forward networks to generate multi-step forecasting (see Section 3.1 for more details). Note that we adopt the frequency-invariant FGSO ($d \times d$) in the EV-FGN where the feed-forward computations act on the embedding dimension, i.e., $d$. In addition, when extending Definition 2 to 2D domain, the frequency-invariant FGSO is invariant to both frequency components derived from the spatial dimension ($N$) and time dimension ($T$) respectively.

### C.3 INTERPRETATION OF OMITTING THE WEIGHT MATRIX $W$ IN PROPOSITION 1

This treatment of omitting the weight matrix $W$ in Eq. 9 is feasible since the weight matrix $W$ can be absorbed in the embedding input, precisely the embedding matrix parameters. Note that we feed the input embeddings into EV-FGN (refer to Fig. 1 and Section 3.1). In addition, this treatment will not reduce the capability of EV-FGN intuitively since we adopt edge-varying filters in EV-FGN. Note that traditional GCNs adopt different weight matrices (regarding the model capability) but the same GSO (e.g., adjacency and Laplacian matrices regarding the given graph structure) in different diffusion orders. Differently, there is no pre-given graph structure in MTS forecasting scenarios, therefore we adopt the edge-varying filters, i.e., varying GSOs in EV-FGN, which does not reduce the model capability and achieves desirable performance empirically.

## D PROOFS

### D.1 PROOF OF PROPOSITION 1

The proof aims to expand the graph convolution corresponding to $H_{EV}$ using a set of FGSOs in the Fourier space. According to the concise form of Equation 7, i.e.,

$$\mathcal{F}(SX) = \mathcal{F}(X) \times_n \mathcal{S} \tag{14}$$

where $\mathcal{F}$ denotes the discrete Fourier transform (DFT) and we omit the weight matrix $W$ for convenience (we can treat $W$ as an identity matrix), it yields:

$$
\begin{aligned}
\mathcal{F}(S_K S_{K-1} \cdots S_0 X) &= \mathcal{F}(S_K(S_{K-1}...S_0 X)) \\
&= \mathcal{F}(S_{K-1}...S_0 X) \times_n \mathcal{S}_K \\
&= \mathcal{F}(\mathcal{X}) \times_n \mathcal{S}_0 \cdots \mathcal{S}_{K-1} \times_n \mathcal{S}_K
\end{aligned}
\tag{15}
$$

where $\{S_i\}_{i=0}^K$ is a set of GSOs, and $\{\mathcal{S}_i\}_{i=0}^K$ is a set of FGSOs corresponding to $\{S_i\}_{i=0}^K$ individually. Thus, the edge-varying graph filter $H_{EV}$ can be rewritten as

$$
\begin{aligned}
\mathcal{F}(H_{EV}X) &= \mathcal{F}(S_0 X + S_1 S_0 X + ... + S_K S_{K-1}...S_0 X) \\
&= \mathcal{F}(S_0 X) + \mathcal{F}(S_1 S_0 X) + ... + \mathcal{F}(S_K S_{K-1}...S_0 X) \\
&= \mathcal{F}(X) \times_n \mathcal{S}_0 + \mathcal{F}(X) \times_n \mathcal{S}_0 \times_n \mathcal{S}_1 + ... + \mathcal{F}(X)\mathcal{S}_0 \times_n \cdots \mathcal{S}_{K-1} \times_n \mathcal{S}_K
\end{aligned}
\tag{16}
$$

Accordingly, we have

$$
H_{EV}X = \mathcal{F}^{-1}\left(\sum_{k=0}^{K} \mathcal{F}(X) \times_n \mathcal{S}_{0:k}\right) \quad s.t. \, \mathcal{S}_{0:k} = \mathcal{S}_0 \times_n \cdots \mathcal{S}_{K-1} \times_n \mathcal{S}_K
\tag{17}
$$

Proved.

## E  COMPARED WITH OTHER NETWORKS

### E.1  GNN

**Graph Convolutional Networks**. Graph convolutional networks (GCNs) depend on the Laplacian eigenbasis to perform the multi-order graph convolutions over a given graph structure. Compared with GCNs, EV-FGN as an efficient alternative to multi-order graph convolutions has three main differences: 1) No eigendecompositions or similar costly matrix operations are required. EV-FGN transforms the input into Fourier domain by discrete Fourier transform (DFT) instead of graph Fourier transform (GFT); 2) Explicitly assigning various importance to nodes of the same neighborhood with different diffusion steps. EV-FGN adopts different Fourier graph shift operators $\mathcal{S}$ in different diffusion steps corresponding to the time-varying dependencies among nodes; 3) EV-FGN is invariant to the discretization $N, T$. It parameterizes the graph convolution via Fourier bases invariant graph structure and graph scale.

**Graph Attention Networks**. Graph attention networks (GATs) are non-spectral attention-based graph neural networks. GATs use node representations to calculate the attention weights (i.e., edge weights) varying with different graph attention layers. Accordingly, both GATs and EV-FGN do not depend on eigendecompositions and adopt varying edge weights with different diffusion steps (layers). However, EV-FGN can efficiently perform graph convolutions in the Fourier space. For a complete graph, the time complexity of the attention calculation of $K$ layers is proportional to $Kn^2$ where $n$ is the number of nodes, while a $K$-layer EV-FGN infers the graph structure in Fourier space with the time complexity proportional to $n \log n$. In addition, compared with GATs that implicitly achieve edge-varying weights with different layers, EV-FGN adopts different FGSOs in different diffusion steps explicitly.

### E.2  FNO

Inspired by Fourier neural operator (FNO) Li et al. (2021) that computes the global convolutions in the Fourier space, we elaborately design EV-FGN to compute the multi-order graph convolutions in the Fourier space. Both FNO and EV-FGN replace the time-consuming graph/global convolutions in time domain with the efficient spectral convolution in the Fourier space according to the convolution theorem.

However, FNO and EV-FGN are quite different in the network architecture. As shown in Fig. 5, FNO consists of a stack of Fourier layers where each Fourier layer serially performs 1) DFT to obtain the spectrum of the input, 2) then the spectral convolution in the Fourier space, 3) and finally IDFT to transform the output to the time domain. Accordingly, FNO needs $K$ pairs of DFT and IDFT for $K$ Fourier layers in FNO. In contrast, EV-FGN with just a pair of DFT and IDFT performs multi-order

(multi-layer) graph convolutions on one fly via stacking multiple FGSOs in Fourier space, as shown in Fig. 1.

In addition, adaptive Fourier neural operator (AFNO) Guibas et al. (2022) is a variant of FNO. It is a neural unit and used in a plug-and-play fashion as an alternative to self-attention to reduce the quadratic complexity. Similarly, it requires a pair of DFT and IDFT to accomplish the self-attention computation. In contrast, our proposed EV-FGN is a neural network with a set of FGSOs in a well-designed connection.

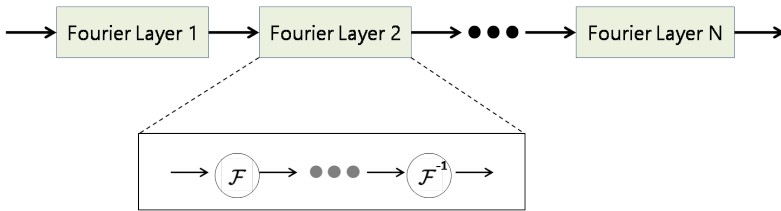

Figure 5: The simplified structure of FNO derived from Li et al. (2021). $\mathcal{F}$ and $\mathcal{F}^{-1}$ denote Fourier transform and its reverse respectively.

## F EXPERIMENT DETAILS

### F.1 DATASETS

**Solar**[1]: This dataset is about the solar power collected by National Renewable Energy Laboratory. We choose the power plant data points in Florida as the data set which contains 593 points. The data is collected from 2006/01/01 to 2016/12/31 with the sampling interval of every 1 hour.

**Wiki**[2]: This dataset contains a number of daily views of different Wikipedia articles and is collected from 2015/7/1 to 2016/12/31. It consists of approximately $145k$ time series and we randomly choose $2k$ from them as our experimental data set.

**Traffic**[3]: This dataset contains hourly traffic data from 963 San Francisco freeway car lanes. The traffic data are collected since 2015/01/01 with the sampling interval of every 1 hour.

**ECG**[4]: This dataset is about Electrocardiogram(ECG) from the UCR time-series classification archive Dau et al. (2019). It contains 140 nodes and each node has a length of 5000.

**Electricity**[5]: This dataset contains electricity consumption of 370 clients and is collected since 2011/01/01. Data sampling interval is every 15 minutes.

**COVID-19**[6]: This dataset is about COVID-19 hospitalization in the U.S. states of California (CA) from 01/02/2020 to 31/12/2020 provided by the Johns Hopkins University with the sampling interval of every one day.

**METR-LA**[7]: This dataset contains traffic information collected from loop detectors in the highway of Los Angeles County. It contains 207 sensors which is from 01/03/2012 to 30/06/2012 and the data sampling interval is every 5 minutes.

---

[1] https://www.nrel.gov/grid/solar-power-data.html
[2] https://www.kaggle.com/c/web-traffic-time-series-forecasting/data
[3] https://archive.ics.uci.edu/ml/datasets/PEMS-SF
[4] http://www.timeseriesclassification.com/description.php?Dataset=ECG5000
[5] https://archive.ics.uci.edu/ml/datasets/ElectricityLoadDiagrams20112014
[6] https://github.com/CSSEGISandData/COVID-19
[7] https://github.com/liyaguang/DCRNN

## F.2 BASELINES

**VAR** Watson (1993): VAR is a classic linear autoregressive model. We use Statsmodels library (`https://www.statsmodels.org`) which is a python package that provides statistical computations to realize the VAR.

**DeepGLO** Sen et al. (2019): DeepGLO models the relationships among variables by matrix factorization and employs a temporal convolution neural network to introduce non-linear relationships. We download the source code from: `https://github.com/rajatsen91/deepglo`. We use the default setting as our experimental settings for wiki, electricity and traffic datasets. For covid datasets, the vertical and horizontal batch size is set to 64, the rank of global model is set to 64, the number of channels is set to [32, 32, 32, 1], and the period is set to 7.

**LSTNet** Lai et al. (2018): LSTNet uses a CNN to capture inter-variable relationships and a RNN to discover long-term patterns. We download the source code from: `https://github.com/laiguokun/LSTNet`. In our experiment, we use the default settings where the CNN hidden units is 100, the kernel size of the CNN layers is 4, the dropout is 0.2, the RNN hidden units is 100, the number of RNN hidden layers is 1, the learning rate is 0.001 and the optimizer is Adam.

**TCN** Bai et al. (2018): TCN is a causal convolution model for regression prediction. We download the source code from: `https://github.com/locuslab/TCN`. We utilize the same configuration as the polyphonic music task exampled in the open source code where the dropout is 0.25, the kernel size is 5, the hidden units is 150, the number of levels is 4 and the optimizer is Adam.

**Reformer** Kitaev et al. (2020): Reformer combines the modeling capacity of a Transformer with an architecture that can be executed efficiently on long sequences and with small memory use. We download the source code from: `https://github.com/thuml/Autoformer`. We use the recommended settings as the experimental settings.

**Informer** Zhou et al. (2021): Informer leverages an efficient self-attention mechanism to encode the dependencies among variables. We download the source code from: `https://github.com/zhouhaoyi/Informer2020`. We use the default settings as our experimental settings where the dropout is 0.05, the number of encoder layers is 2, the number of decoder layers is 1, the learning rate is 0.0001, and the optimizer is Adam.

**Autoformer** Wu et al. (2021): Autoformer proposes a decomposition architecture by embedding the series decomposition block as an inner operator, which can progressively aggregate the long-term trend part from intermediate prediction. We download the source code from: `https://github.com/thuml/Autoformer`. We use the recommended settings as our experimental settings with 2 encoder layers and 1 decoder layer.

**FEDformer** Zhou et al. (2022): FEDformer proposes an attention mechanism with low-rank approximation in frequency and a mixture of experts decomposition to control the distribution shifting. We download the source code from: `https://github.com/MAZiqing/FEDformer`. We use FEB-f as the Frequency Enhanced Block and select the random mode with 64 as the experimental mode.

**SFM** Zhang et al. (2017): On the basis of the LSTM model, SFM introduces a series of different frequency components in the cell states. We download the source code from: `https://github.com/z331565360/State-Frequency-Memory-stock-prediction`. We use the default settings as the authors recommended where the learning rate is 0.01, the frequency dimension is 10, the hidden dimension is 10 and the optimizer is RMSProp.

**StemGNN** Cao et al. (2020): StemGNN leverages GFT and DFT to capture dependencies among variables in the frequency domain. We download the source code from: `https://github.com/microsoft/StemGNN`. We use the default setting of stemGNN as our experiment setting where the optimizer is RMSProp, the learning rate is 0.0001, the stacked layers is 5, and the dropout rate is 0.5.

**MTGNN** Wu et al. (2020): MTGNN proposes an effective method to exploit the inherent dependency relationships among multiple time series. We download the source code from: `https://github.com/nnzhan/MTGNN`. Because the experimental datasets have no static features, we set the

parameter load_static_feature to false. We construct the graph by the adaptive adjacency matrix and add the graph convolution layer. Regarding other parameters, we adopt the default settings.

**GraphWaveNet** Wu et al. (2019): GraphWaveNet introduces an adaptive dependency matrix learning to capture the hidden spatial dependency. We download the source code from: `https://github.com/nnzhan/Graph-WaveNet`. Since our datasets have no prior defined graph structures, we use only adaptive adjacent matrix. We add a graph convolution layer and randomly initialize the adjacent matrix. We adopt the default setting as our experimental settings where the learning rate is 0.001, the dropout is 0.3, the number of epoch is 50, and the optimizer is Adam.

**AGCRN** Bai et al. (2020): AGCRN proposes a data-adaptive graph generation module for discovering spatial correlations from data. We download the source code from: `https://github.com/LeiBAI/AGCRN`. We use the default settings as our experimental settings where the embedding dimension is 10, learning rate is 0.003, and the optimizer is Adam.

**TAMP-S2GCNets** Chen et al. (2022): TAMP-S2GCNets explores the utility of MP to enhance knowledge representation mechanisms within the time-aware DL paradigm. We download the source code from: `https://www.dropbox.com/sh/n0ajd5l0tdeyb80/AABGn-ejfV1YtRwjf_L0AOsNa?dl=0`. TAMP-S2GCNets requires predefined graph topology and we use the California State topology provided by the source code as input. We adopt the default settings as our experimental settings on COVID-19.

**DCRNN** Li et al. (2018): DCRNN uses bidirectional graph random walk to model spatial dependency and recurrent neural network to capture the temporal dynamics. We download the source code from : `https://github.com/liyaguang/DCRNN`. We use the default settings as our experimental settings with the batch size is 64, the learning rate is 0.01, the input dimension is 2 and the optimizer is Adam. DCRNN repuires a pre-defined graph structures and we use the adjacency matrix as the pre-defined structures provided by METR-LA dataset.

**STGCN** Yu et al. (2018a): STGCN integrates graph convolution and gated temporal convolution through spatial-temporal convolutional blocks. We download the source code from:`https://github.com/VeritasYin/STGCN_IJCAI-18`. We use the default settings as our experimental settings where the batch size is 50, the learning rate is 0.001 and the optimizer is Adam. STGCN requires a pre-defined graph structures and we leverage the adjacency matrix as the pre-defined structures provided by METR-LA dataset.

**CoST** Woo et al. (2022): CoST separates the representation learning and downstream forecasting task and proposes a contrastive learning framework that learns disentangled season-trend representations for time series forecasting tasks. We download the source code from: `https://github.com/salesforce/CoST`. We set the representation dimension to 320, a learning rate to 0.001 and the batch size to 32. Inputs are min-max normalization, we perform a 70/20/10 train/validation/test split for METR-LA dataset and 60/20/20 for COVID-19 dataset.

### F.3 EVALUATION METRICS

We use MAE (Mean Absolute Error), RMSE (Root Mean Square Error), and MAPE (Mean Absolute Percentage Error) as the evaluation metrics in the experiments.

Specifically, given the groudtruth $X^{t+1:t+\tau} \in \mathbb{R}^{N \times \tau}$ and the predictions $\hat{X}^{t+1:t+\tau} \in \mathbb{R}^{N \times \tau}$ for future $\tau$ steps at timestamp $t$, the metrics are defined as follows:

$$MAE = \frac{1}{\tau N} \sum_{i=1}^{N} \sum_{j=1}^{\tau} |x_{ij} - \hat{x}_{ij}| \tag{18}$$

$$RMSE = \sqrt{\frac{1}{\tau N} \sum_{i=1}^{N} \sum_{j=1}^{\tau} (x_{ij} - \hat{x}_{ij})^2} \tag{19}$$

$$MAPE = \frac{1}{\tau N} \sum_{i=1}^{N} \sum_{j=1}^{\tau} \left| \frac{x_{ij} - \hat{x}_{ij}}{x_{ij}} \right| \times 100\% \tag{20}$$

with $x_{ij} \in X^{t+1:t+\tau}$ and $\hat{x}_{ij} \in \hat{X}^{t+1:t+\tau}$.

F.4    EXPERIMENTAL SETTINGS

We summarize the implementation details of the proposed EV-FGN as follows. Note that the details of the baselines are introduced in their corresponding descriptions (see Section F.2).

**Network details.** The fully connected feed-forward network (FFN) consists of three linear transformations with $LeakyReLU$ activations in between. The FFN is formulated as follows:

$$
\begin{aligned}
\mathbf{X}_1 &= \text{LeakyReLU}(\mathbf{X}_\Psi \mathbf{W}_1 + \mathbf{b}_1) \\
\mathbf{X}_2 &= \text{LeakyReLU}(\mathbf{X}_1 \mathbf{W}_2 + \mathbf{b}_2) \\
\hat{X} &= \mathbf{X}_2 \mathbf{W}_3 + \mathbf{b}_3
\end{aligned}
\tag{21}
$$

where $\mathbf{W}_1 \in \mathbb{R}^{(Td) \times d_1^{ffn}}$, $\mathbf{W}_2 \in \mathbb{R}^{d_1^{ffn} \times d_2^{ffn}}$ and $\mathbf{W}_3 \in \mathbb{R}^{d_2^{ffn} \times \tau}$ are the weights of the three layers respectively, and $\mathbf{b}_1 \in \mathbb{R}^{d_1^{ffn}}$, $\mathbf{b}_2 \in \mathbb{R}^{d_2^{ffn}}$ and $\mathbf{b}_3 \in \mathbb{R}^{\tau}$ are the biases of the three layers respectively. Here, $d_1^{ffn}$ and $d_2^{ffn}$ are the dimensions of the three layers. In addition, we adopt a $ReLU$ activation function in Equation 11.

Table 6: Dimension settings of FFN on different datasets.

| Datasets | Solar | Wiki | Traffic | ECG | Electricity | COVID-19 | META-LR |
|---|---|---|---|---|---|---|---|
| $l$ | 6 | 2 | 2 | $*$ | 4 | 8 | 4 |
| $d_1^{ffn}$ | 64 | 64 | 64 | 64 | 64 | 256 | 64 |
| $d_2^{ffn}$ | 256 | 256 | 256 | 256 | 256 | 512 | 256 |

$*$ denotes that we feed the original time domain representation to FFN without the dimension reduction.

**Training details.** We carefully tune the hyperparameters, including the embedding size, batch size, $d_1^{ffn}$ and $d_2^{ffn}$, on the validation set and choose the settings with the best performance for EV-FGN on different datasets. Specifically, the embedding size and batch size are tuned over $\{32, 64, 128, 256, 512\}$ and $\{2, 4, 8, 16, 32, 64, 128\}$ respectively. For the COVID-19 dataset, the embedding size is 256, and the batch size is set to 4. For the Traffic, Solar and Wiki datasets, the embedding size is 128, and the batch size is set to 2. For the METR-LA, ECG and Electricity datasets, the embedding size is 128, and the batch size is set to 32. Note that the supra-graph connecting all nodes is a fully-connected graph, indicating that any spatial order is feasible. Therefore, although we perform 2D DFT in EV-FGN, none spatial order in the spatial space is necessary for performing EV-FGN, and we directly adopt the raw dataset for experiments.

To reduce the number of parameters, we adopt a linear transform to map the original time domain representation $\mathbf{X}_\Psi \in \mathbb{R}^{N \times T \times d}$ to a low-dimensional tensor $\mathbf{X}_\Psi \in \mathbb{R}^{N \times l \times d}$ with $l < T$. We then reshape $\mathbf{X}_\Psi \in \mathbb{R}^{N \times (ld)}$ and feed it to FFN. We perform grid search on the dimensions of FFN, i.e., $d_1^{ffn}$ and $d_2^{ffn}$, over $\{32, 64, 128, 256, 512\}$ and tune the intermediate dimension $l$ over $\{2, 4, 6, 8, 12\}$. The settings of the three hyperparameters over all datasets are shown in Table 6. Finally, we set the diffusion step $K = 3$ for all datasets.

F.5    DETAILS FOR VISUALIZATION EXPERIMENTS

To verify the effectiveness of EV-FGN in learning the spatial-temporal dependencies on the fully-connected supra-graph, we obtain the output of EV-FGN as the node representation, denoted as $\mathbf{R} = \text{IDFT}(\Psi_K(\mathcal{X})) \in \mathbb{R}^{N \times T \times d}$ with inverse discrete Fourier transform (IDFT) and $K$-layer EV-FGN $\Psi_K$. Then, we visualize the adjacency matrix $\mathbf{A}$ calculated based the flatten node representation $\mathbf{R} \in \mathbb{R}^{NT \times d}$, formulated as $\mathbf{A} = \mathbf{R}\mathbf{R}^T \in \mathbb{R}^{NT \times NT}$, to show the variable correlations. Note that $\mathbf{A}$ is normalized via $\mathbf{A}/\max(\mathbf{A})$. Since it is not feasible to directly visualize the huge adjacency matrix $\mathbf{A}$ of the supra-graph, we visualize its different subgraphs in Figures 3, 4, 5, and 10 to better verify the learned spatial-temporal information on the supra-graph from different perspectives.

Figure 3: On the METR-LA dataset, we average its adjacency matrix $\mathbf{A}$ over the temporal dimension (i.e., marginalizing $T$) to $\mathbf{A}' \in \mathbb{R}^{N \times N}$. Then, we randomly select 20 detectors out of all $N = 207$ detectors and obtain their corresponding sub adjacency matrix ($\mathbb{R}^{20 \times 20}$) from $\mathbf{A}'$ for visualization.

We further compare the sub adjacency with the real road map (generated by the google map tool) to verify the learned dependencies between different detectors.

Figure 4. On the COVID-19 dataset, we randomly choose 10 counties out of $N = 55$ counties and obtain their four sub adjacency matrices of four consecutive days for visualization. Each of the four sub adjacency matrices $\mathbb{R}^{10 \times 10}$ embodies the dependencies between counties in one day. Figure 4 reflects the time-varying dependencies between counties (i.e., variables).

Figure 5. Since we adopt a 3-layer EV-FGN, we can calculate four adjacency matrices based on the input $\mathcal{X}$ of EV-FGN and the outputs of each layer in EV-FGN, i.e., $\mathcal{X}_1, \mathcal{X}_2, \mathcal{X}_3$. Following the way of visualization in Figure 4, we select 10 counties and two timestamps on the four adjacency matrices for visualization. Figure 5 shows the effects of each layer of EV-FGN in filtering or enhancing variable correlations.

Figure 10. We select 8 counties and visualize the correlations between 12 consecutive time steps for each selected county respectively. Figure 10 reflects the temporal correlations within each variable.

## G   MORE RESULTS

Table 7: Performance comparison under different prediction lengths on the COVID-19 dataset.

| Length | 3 | | | 6 | | | 9 | | | 12 | | |
|---|---|---|---|---|---|---|---|---|---|---|---|---|
| Metrics | MAE | RMSE | MAPE(%) | MAE | RMSE | MAPE(%) | MAE | RMSE | MAPE(%) | MAE | RMSE | MAPE(%) |
| GraphWaveNet Wu et al. (2019) | 0.092 | 0.129 | 53.00 | 0.133 | 0.179 | 65.11 | 0.171 | 0.225 | 80.91 | 0.201 | 0.255 | 100.83 |
| StemGNN Cao et al. (2020) | 0.247 | 0.318 | 99.98 | 0.344 | 0.429 | 125.81 | 0.359 | 0.442 | 131.14 | 0.421 | 0.508 | 141.01 |
| AGCRN Bai et al. (2020) | 0.130 | 0.172 | 68.64 | 0.171 | 0.218 | 79.29 | 0.224 | 0.277 | 113.42 | 0.254 | 0.309 | 125.43 |
| MTGNN Wu et al. (2020) | 0.276 | 0.379 | 91.42 | 0.446 | 0.513 | 133.49 | 0.484 | 0.548 | 139.52 | 0.394 | 0.488 | 88.13 |
| TAMP-S2GCNets Chen et al. (2022) | 0.140 | 0.190 | **50.01** | 0.150 | 0.200 | **55.72** | 0.170 | 0.230 | 71.78 | 0.180 | 0.230 | **65.76** |
| CoST Woo et al. (2022) | 0.122 | 0.246 | 68.74 | 0.157 | 0.318 | 72.84 | 0.183 | 0.364 | 77.04 | 0.202 | 0.377 | 80.81 |
| **EV-FGN(ours)** | **0.071** | **0.103** | 61.02 | **0.093** | **0.131** | 65.72 | **0.109** | **0.148** | 69.59 | **0.124** | **0.164** | 72.57 |
| Improvement | 22.8% | 20.2% | - | 30.1% | 26.8% | - | 35.9% | 35.7% | 3.1 % | 31.1% | 28.7% | - |

To further evaluate the performance of our model EV-FGN in multi-step forecasting, we conduct more experiments on the COVID-19, Wiki, ECG, and METR-LA datasets, respectively. Note that the **COVID-19** and **METR-LA** datasets have predefined graph topologies. We compare EV-FGN with other GNN-based MTS models (including StemGNN, AGCRN, GraphWaveNet, MTGNN and TAMP-S2GCNets) and representation learning model (CoST) on the COVID-19 dataset under different prediction lengths, and the results are shown in Table 7. From the table, we can find that EV-FGN achieves the best MAE and RMSE on all the prediction lengths. On average, EV-FGN has 30.0% and 27.9% improvement on MAE and RMSE respectively over the best baseline, i.e., TAMP-S2GCNets. Among these models, TAMP-S2GCNets requiring a pre-defined graph topology achieves competitive performance since it enhances the resultant graph learning mechanisms with a multi-persistence. However, it constructs the graph in the spatial dimension, while our model EV-FGN adaptively learns a supra-graph connecting any two variables at any two timestamps, which is effective and more powerful to capture high-resolution spatial-temporal dependencies.

Table 8: Performance comparison under different prediction lengths on the Wiki dataset.

| Length | 3 | | | 6 | | | 9 | | | 12 | | |
|---|---|---|---|---|---|---|---|---|---|---|---|---|
| Metrics | MAE | RMSE | MAPE(%) | MAE | RMSE | MAPE(%) | MAE | RMSE | MAPE(%) | MAE | RMSE | MAPE(%) |
| GraphWaveNet Wu et al. (2019) | 0.061 | 0.105 | 138.60 | 0.061 | 0.105 | 135.32 | 0.061 | 0.105 | 132.52 | 0.061 | 0.104 | 136.12 |
| StemGNN Cao et al. (2020) | 0.157 | 0.236 | 89.00 | 0.159 | 0.233 | 98.01 | 0.232 | 0.311 | 142.14 | 0.220 | 0.306 | 125.40 |
| AGCRN Bai et al. (2020) | 0.043 | 0.077 | 73.49 | 0.044 | 0.078 | 80.44 | 0.045 | 0.079 | 81.89 | 0.044 | 0.079 | 78.52 |
| MTGNN Wu et al. (2020) | 0.102 | 0.141 | 123.15 | 0.091 | 0.133 | 91.75 | 0.074 | 0.120 | 85.44 | 0.101 | 0.140 | 122.96 |
| Informer Zhou et al. (2021) | 0.053 | 0.089 | 85.31 | 0.054 | 0.090 | 84.46 | 0.059 | 0.095 | 93.80 | 0.059 | 0.095 | 95.09 |
| **EV-FGN(ours)** | **0.040** | **0.075** | **58.18** | **0.041** | **0.075** | **60.43** | **0.041** | **0.076** | **60.95** | **0.042** | **0.077** | **62.62** |
| Improvement | 7.0% | 2.6% | 20.83% | 6.8% | 3.8% | 24.9% | 8.9% | 3.8% | 25.6% | 4.5% | 2.5% | 20.3% |

In addition, we compare our model EV-FGN with five neural MTS models (including StemGNN, AGCRN, GraphWaveNet, MTGNN and Informer) on Wiki dataset under different prediction lengths,

and the results are shown in Table 8. From the table, we observe that EV-FGN outperforms other models on MAE, RMSE and MAPE metrics for all the prediction lengths. On average, EV-FGN improves MAE, RMSE and MAPE by 6.8%, 3.2% and 22.9%, respectively. Among these models, AGCRN shows promising performances since it captures the spatial and temporal correlations adaptively. However, it fails to simultaneously capture spatial-temporal dependencies, limiting its forecasting performance. In contrast, our model learns a supra-graph to capture comprehensive spatial-temporal dependencies simultaneously for multivariate time series forecasting.

Table 9: Performance comparison under different prediction lengths on the METR-LA dataset.

| Horizon | 3 | | | 6 | | | 9 | | | 12 | | |
|---|---|---|---|---|---|---|---|---|---|---|---|---|
| Metrics | MAE | RMSE | MAPE(%) | MAE | RMSE | MAPE(%) | MAE | RMSE | MAPE(%) | MAE | RMSE | MAPE(%) |
| DCRNN Li et al. (2018) | 0.160 | 0.204 | 80.00 | 0.191 | 0.243 | 83.15 | 0.216 | 0.269 | 85.72 | 0.241 | 0.291 | 88.25 |
| STGCN Yu et al. (2018a) | 0.058 | 0.133 | 59.02 | 0.080 | 0.177 | 60.67 | 0.102 | 0.209 | 62.08 | 0.128 | 0.238 | 63.81 |
| GraphWaveNet Wu et al. (2019) | 0.180 | 0.366 | 21.90 | 0.184 | 0.375 | 22.95 | 0.196 | 0.382 | 23.61 | 0.202 | 0.386 | 24.14 |
| MTGNN Wu et al. (2020) | 0.135 | 0.294 | **17.99** | 0.144 | 0.307 | **18.82** | 0.149 | 0.328 | **19.38** | 0.153 | 0.316 | **19.92** |
| StemGNN Cao et al. (2020) | 0.052 | 0.115 | 86.39 | 0.069 | 0.141 | 87.71 | 0.080 | 0.162 | 89.00 | 0.093 | 0.175 | 90.25 |
| AGCRN Bai et al. (2020) | 0.062 | 0.131 | 24.96 | 0.086 | 0.165 | 27.62 | 0.099 | 0.188 | 29.72 | 0.109 | 0.204 | 31.73 |
| Informer Zhou et al. (2021) | 0.076 | 0.141 | 69.96 | 0.088 | 0.163 | 70.94 | 0.096 | 0.178 | 72.26 | 0.100 | 0.190 | 72.54 |
| CoST Woo et al. (2022) | 0.064 | 0.118 | 88.44 | 0.077 | 0.141 | 89.63 | 0.088 | **0.159** | 90.56 | 0.097 | 0.171 | 91.42 |
| **EV-FGN(ours)** | **0.050** | **0.113** | 86.30 | **0.066** | **0.140** | 87.97 | **0.076** | **0.159** | 88.99 | **0.084** | **0.165** | 89.69 |
| Improvement | 3.8% | 1.7% | - | 4.3% | 0.7% | - | 5.0% | - | - | 9.7% | 3.5% | - |

Finally, we compare our model EV-FGN with seven neural MTS models (including STGCN, DCRNN, StemGNN, AGCRN, GraphWaveNet, MTGNN, Informer and CoST) on the METR-LA dataset, and the results are shown in Table 9. On average, we improve 5.7% on MAE and 2.5% on RMSE. Among these models, StemGNN achieves competitive performance because it combines GFT to capture the spatial dependencies and DFT to capture the temporal dependencies. However, it is also limited to simultaneously capturing spatial-temporal dependencies. CoST learns disentangled seasonal-trend representations for time series forecasting via contrastive learning and obtains competitive results. But, our model still outperforms CoST. Because, compared with CoST, our model not only can learn the dynamic temporal representations, but also capture the discriminative spatial representations. Besides, STGCN and DCRNN require pre-defined graph structures. But StemGNN and our model outperform them for all steps, and AGCRN outperforms them when the prediction lengths are 9 and 12. This also shows that a novel adaptive graph learning can precisely capture the hidden spatial dependency. In addition, we compare EV-FGN with the baseline models under the different prediction lengths on the ECG dataset, as shown in Fig. 6. It reports that EV-FGN achieves the best performances (MAE, RMSE and MAPE) for all prediction lengths.

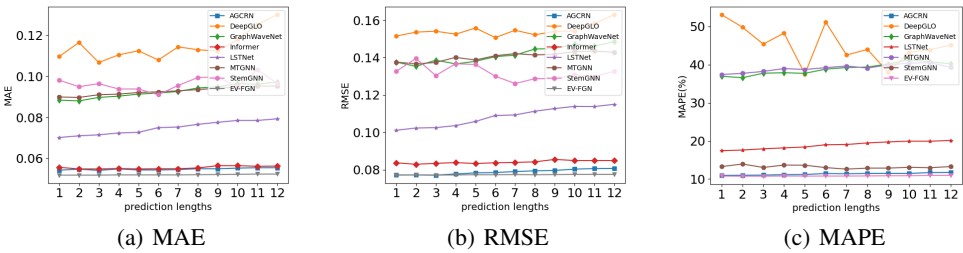

|  (a) MAE | (b) RMSE | (c) MAPE |
|---|---|---|

Figure 6: Performance comparison in different prediction lengths on the ECG dataset.

# H  MORE ANALYSES

## H.1  ANALYSES

**Parameter Analysis**. We evaluate the forecasting performance of our model EV-FGN under different diffusion steps on the COVID-19 dataset, as illustrated in Table 10. The table shows that EV-FGN achieves increasingly better performance from $K = 1$ to $K = 4$ and achieves the best results when $K = 3$. With the further increase of $K$, EV-FGN obtains inferior performance. The results indicate that high-order diffusion information is beneficial for improving the forecasting accuracy, but the

Table 10: Performance at different diffusion steps on the COVID-19 dataset.

|         | K=1   | K=2   | K=3      | K=4   |
|---------|-------|-------|----------|-------|
| MAE     | 0.136 | 0.133 | 0.129 | 0.132 |
| RMSE    | 0.181 | 0.177 | 0.173 | 0.176 |
| MAPE(%) | 72.30 | 71.80 | 71.52 | 72.59 |

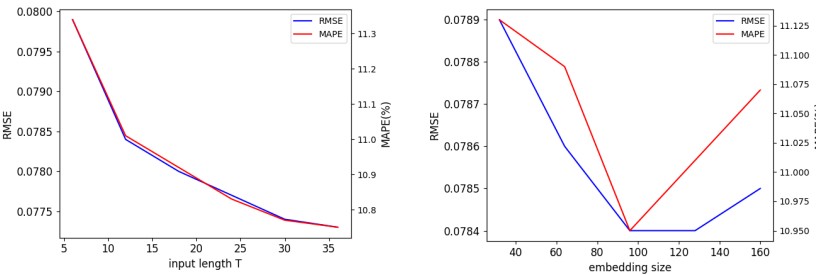

Figure 7: Influence of input length.    Figure 8: Influence of embedding size.

diffusion information may gradually weaken the effect or even bring noises to forecasting with the increase of the order.

In addition, we conduct extensive experiments on the ECG dataset to analyze the effect of the input length and the embedding dimension $d$, as shown in Fig. 7 and Fig. 8, respectively. Fig. 7 shows that the performance (including RMSE and MAPE) of EV-FGN gets better as the input length increases, indicating that EV-FGN can learn a comprehensive supra-graph from long MTS inputs to capture the spatial and temporal dependencies. Moreover, Fig. 8 shows that the performance (RMSE and MAPE) first increases and then decreases with the increase of the embedding size, which is attributed that a large embedding size improves the fitting ability of EV-FGN but it may easily lead to the overfitting issue especially when the embedding size is too large.

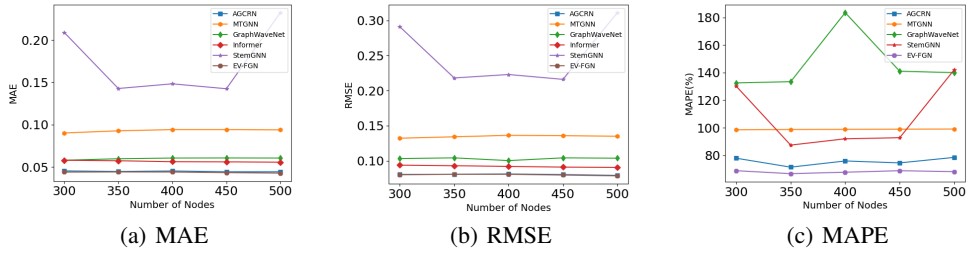

(a) MAE    (b) RMSE    (c) MAPE

Figure 9: The sensitivity between MAE, RMSE, MAPE and number of nodes on the Wiki dataset.

We further conduct experiments on the Wiki dataset to investigate the performance of EV-FGN under different graph sizes. The results are shown in Fig. 9, where Fig. 9(a), Fig. 9(b) and Fig. 9(c) show MAE, RMSE and MAPE at the different number of nodes, respectively. From these figures, we observe that EV-FGN keeps a leading edge over the other state-of-the-art MTS models as the number of nodes increases. The results demonstrate the superiority and scalability of EV-FGN on large scale datasets.

**Ablation Study**. We provide more details about each variant used in this section and Section 4.3.

- **w/o Embedding**. A variant of EV-FGN feeds the raw MTS input instead of its embeddings into the graph convolution in the Fourier space.

Table 11: Ablation studies on the COVID-19 dataset.

| metrics | w/o Embedding | w/o Dynamic Filter | w/o Residual | w/o Summation | EV-FGN |
|---------|---------------|--------------------|--------------|--------------|---------|
| MAE | 0.157 | 0.138 | 0.131 | 0.134 | 0.129 |
| RMSE | 0.203 | 0.180 | 0.174 | 0.177 | 0.173 |
| MAPE(%) | 76.91 | 74.01 | 72.25 | 72.57 | 71.52 |

- **w/o Dynamic Filter**. A variant of EV-FGN uses the same FGSO for all diffusion steps instead of applying different FGSOs in different diffusion steps. It corresponds to a vanilla graph filter.

- **w/o Residual**. A variant of EV-FGN does not have the $K = 0$ layer output, i.e., $\mathcal{X}$, in the summation.

- **w/o Summation**. A variant of EV-FGN adopts the last order (layer) output $\mathcal{X}_K$ as the final frequency output of the EV-FGN.

We conduct another ablation study on the COVID-19 dataset to further investigate the effects of the different components of our EV-FGN. The results are shown in Table 11, which confirms the results in Table 4 and further verifies the effectiveness of each component in EV-FGN. Both Table 11 and Table 4 report that the embedding and dynamic filter in EV-FGN contribute more than the design of residual and summation to the state-of-the-art performance of EV-FGN.

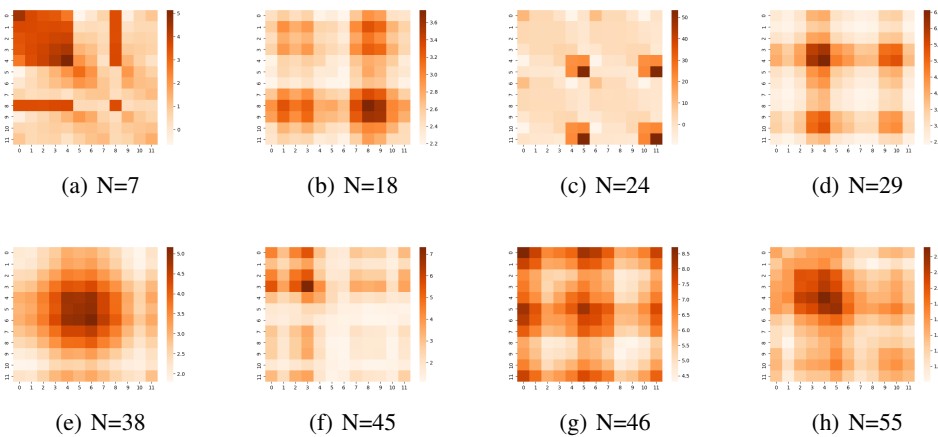

| (a) N=7 | (b) N=18 | (c) N=24 | (d) N=29 |
|---------|----------|----------|----------|

| (e) N=38 | (f) N=45 | (g) N=46 | (h) N=55 |
|----------|----------|----------|----------|

Figure 10: The temporal adjacency matrix of eight variables on COVID-19 dataset.

**Ordering of the time series**. Note that the supra-graph connecting all nodes is a fully-connected graph, indicating that any spatial order is feasible. Then how could we perform 2D DFT to achieve the graph convolution? First, the graph convolution on the supra-graph can be viewed as a kernel summation (cf. Eq.4, i.e., $O(X)[i] = \sum_{j=1}^{n} X[j]\kappa[i,j], \forall j \in [n]$) and does not depend on a specific spatial order. From Eq.4 to Eq.5, we extend the kernel summation to a kernel integral in the continuous spatial-temporal space and introduce a special kernel, i.e., the shift-invariant Green's kernel $\kappa(i,j) = \kappa(i - j)$. According to the convolution theorem, we can reformulate the graph convolution with continuous Fourier transform (i.e., Eq.6 $\mathcal{O}(X)(i) = \mathcal{F}^{-1}\left(\mathcal{F}(X)\mathcal{F}(\kappa)\right)(i), \forall i \in \mathcal{D})$ and then apply Eq.6 to the finite discrete spatial-temporal space (i.e., the supra-graph). Accordingly, we can obtain the Definition 2 and Eq.7 reformulating the graph convolution with 2D DFT.

To understand the learning paradigm of EV-FGN, we can recall the self-attention mechanism as an analogy that does not need any assumption on datapoint order and introduce extra position embeddings for capturing temporal patterns. EV-FGN performs 2D DFT on each discrete $N \times T$ spatial-temporal plane of the embeddings $\mathbf{X}$ instead of the raw data $X$. A key insight underpinning EV-FGN is to introduce variable embeddings and temporal position embeddings to equip EV-FGN with a sense of variable correlations (spatial patterns and temporal patterns).

To verify the claim, we conducted experiments on the dataset ECG. Specifically, we randomly shuffle the order of time-series variables of the raw data five times and evaluate our EV-FGN on each shuffled data. The results are reported in the following table. Note that we can also conduct shuffling experiments on temporal order via performing the same shuffling scheme over each window-sized time-series input, which will get the same conclusion.

Table 12: Five-round results on randomly shuffled data on the dataset ECG.

| Metric | R1 | R2 | R3 | R4 | R5 | Raw |
|---|---|---|---|---|---|---|
| MAE | 0.052 | 0.053 | 0.053 | 0.053 | 0.052 | 0.052 |
| RMSE | 0.078 | 0.078 | 0.078 | 0.078 | 0.078 | 0.078 |
| MAPE(%) | 10.95 | 10.98 | 11.02 | 10.99 | 10.99 | 11.05 |

## H.2 VISUALIZATIONS

To demonstrate the ability of our EV-FGN in jointly learning spatial-temporal dependencies, we visualize the temporal adjacency matrix of different variables. Note that the spatial adjacency matrices of different days are reported in Fig. 3. Specifically, we randomly select 8 counties from the COVID-19 dataset and calculate the correlations of 12 consecutive time steps for each county. Then we visualize the adjacency matrix via heat map, and the results are shown in Fig. 10 where $N$ denotes the index of the country (variable). From the figure, we observe that EV-FGN learns clear and specific temporal patterns for each county. These results show that our EV-FGN can not only learn highly interpretable spatial correlations (see Fig. 2 and Fig. 3), but also capture discriminative temporal patterns.

