# OpenReview forum: "Edge-Varying Fourier Graph Network for Multivariate Time Series Forecasting"
_ICLR.cc/2023/Conference — Submitted to ICLR 2023_

### Official Review · Reviewer_cY1E · 2022-10-25

**Confidence:** 3
**Correctness:** 3
**Technical Novelty And Significance:** 2
**Empirical Novelty And Significance:** 2
**Recommendation:** 5

**Clarity, Quality, Novelty And Reproducibility:**


I have provided detailed comments related to clarity, quality, novelty, and reproducibility in the weaknesses section.


**Strength And Weaknesses:**

Strengthes:
1. This paper addresses the ever-changing correlations over time-series data, which is practical and complement previous works.
2. This paper presents extensive and strong baselines in time series forecasting scenarios. Experimental results demonstrate the effectiveness of the proposed method.


Weaknesses:
1. The author claims this is the first work that designs a complex-valued feed-forward network in the Fourier space for efficiently computing multi-layer graph convolutions. Actually, there are some pioneer works that discuss the relation of spectral space and GNN [1,2], e.g., I wonder if the author only applies their conclusion to the multivariate time series forecasting scenarios or represents more analysis?
2. This paper is  not easy to understand. In section 3.2, the author introduces the notation of the graph shift operator. 1) Generally, there are three types of graph shift, graph size, node feature and graph structure.  This work considers the whole three graph shift of some of it? 2) How does edge-varying connect to multivariate time series forecasting problems?   3) How does the spectral GCN address the corresponding shifted problem? Are there some motivational examples or intuitions here?


[1] Zhu, H. and Koniusz, P., 2020, September. Simple spectral graph convolution. In the International Conference on Learning Representations.
[2] Zhang, S., Tong, H., Xu, J. and Maciejewski, R., 2019. Graph convolutional networks: a comprehensive review. Computational Social Networks, 6(1), pp.1-23.

**Summary Of The Paper:**

This work focuses on multivariate time series forecasting problems. To capture the  ever-changing correlations over time series data, this work defines a Fourier graph shift operator and constructs the efficient edge-varying Fourier graph networks to formulate spatial temporal dependencies for the varying data. Extensive experimental results demonstrate the effectiveness of the proposed method.



**Summary Of The Review:**

This paper studies an interesting research topic and connects multivariate time series forecasting with spectral GCN.. But it has the following limitations:
1. The presentation of this paper is not clear. It is not easy to extract the main idea and intuition.
2.  The contribution of this paper is not clear. It is hard to tell if the author only borrows some existing conclusion in spectral GCN to time series forecasting problems or present some new finding.

---

> ### Author Response · Authors · 2022-11-09
> **Response to Reviewer cY1E (1/2)**
>
> Dear Reviewer cY1E,
>
> thanks for your valuable comments to further improve our manuscripts. We will elaborately address your concerns as follows and carefully revise our manuscript to clarify our contribution and novelty.
>
> **Comments 1:**
>
> The author claims this is the first work that designs a complex-valued feed-forward network in the Fourier space for efficiently computing multi-layer graph convolutions. Actually, there are some pioneer works that discuss the relation of spectral space and GNN [1,2], e.g., I wonder if the author only applies their conclusion to the multivariate time series forecasting scenarios or represents more analysis?
>
> **Response:**
>
> **Our proposed EV-FGN is distinct from the spectral-based GCN**
>
> Note that GCNs generally are divided into two main streams, i.e., the **spectral-based** approaches and the **spatial-based** approaches [wu2021, Zhang2019]. The spectral-based approaches [Zhu2020] are based on Eigen decomposition of the Laplacian matrix of the graph derived from **graph Fourier transform (GFT)**, while the spatial-based approaches [Hamilton2017, Chen2018] earlier than the spectral-based approaches work on the local neighborhood of nodes and understand the properties of a node based on its local neighbors. **Our proposed EV-FGN belongs to the spatial-based approaches**, and no eigendecompositions or similar costly matrix operations are required in the proposed EV-FGN. (please refer to the *Definition 1: Graph Shift Operator*).
>
> More importantly, we reformulate (spatial-based) graph convolutions in the time domain leveraging **discrete Fourier transform (DFT)** in the frequency domain according to the Convolution Theorem [Katznelson1970] (please refer to Eqn. 3~7). *Convolution theorem states that the Fourier transform of a convolution of two functions (or signals) equals the pointwise product of their Fourier transforms*. This treatment can effectively accelerate the time-consuming graph convolutions in the time domain. Accordingly, our EV-FGN adopts DFT to transform inputs (MTS embeddings) into the frequency domain (**complex-valued** intermediate results), while the spectral-based GCNs adopt GFT to decompose the Laplacian matrix (real-valued intermediate results).
>
> In summary, we can conclude the difference between our EV-FGN and the spectral-based GCNs:
> + Theoretically: EV-FGN is based on the Convolution Theorem, while the spectral-based GCNs are based on spectral decomposition.
> + Methodology: EV-FGN adopts DFT to transform inputs, while the spectral-based GCNs adopt GFT to decompose the Laplacian matrix.
> + Architecture: EV-FGN adopts a complex-valued network, while spectral-based GCNs employ real-valued networks.
>
> **The complex-valued feed-forward network is novel and original to efficiently compute multi-layer graph convolutions**
>
> As shown in Eqs. 7 and 8, the multiplication in the frequency domain is a complex-valued matrix multiplication of the spectrums. According to the Convolution theorem, the multiplication of spectrums containing the imaginary part and real part is necessary to achieve the graph/global convolutions. Each layer of the complex-valued feed-forward network corresponds to one graph convolutional layer. To the best of our knowledge, EV-FGN makes the **first** attempt to reformulate graph convolution in the time domain with DFT in the frequency domain and to design a complex-valued feed-forward network to achieve efficient multi-layer graph convolutions.
>
> [wu2021] A Comprehensive Survey on Graph Neural Networks, TNNLS 2021.
>
> [Zhang2019] Graph convolutional networks: a comprehensive review. Computational Social Networks, 2019, 6(1), pp.1-23.
>
> [Zhu2020] September. Simple spectral graph convolution. ICLR 2020.
>
> [Hamilton2017] Inductive representation learning on large graphs, NIPS, 2017.
>
> [Chen2018] Fastgcn: fast learning with graph convolutional networks via importance sampling, ICLR, 2018.
>
> [Katznelson1970] An Introduction to Harmonic Analysis, Cambridge University Press.

---

> ### Author Response · Authors · 2022-11-09
> **Response to Reviewer cY1E (2/2)**
>
> **Comments 2:**
>
> This paper is not easy to understand. In section 3.2, the author introduces the notation of the graph shift operator. 1) Generally, there are three types of graph shift, graph size, node feature and graph structure. This work considers the whole three graph shift of some of it? 2) How does edge-varying connect to multivariate time series forecasting problems? 3) How does the spectral GCN address the corresponding shifted problem? Are there some motivational examples or intuitions here?
>
> **Response:**
>
> 1) Graph shift operator is defined as a general family of operators which enable the diffusion of information over graph structures in signal processing.
> As shown in Definition 1, the graph shift operator (GSO) expresses the graph structure information, i.e., the connection structure in the graph. GSO includes the *adjacency, Laplacian matrices, and their normalizations* as instances of its class. GSO is introduced to define a general form of spatial-based graph convolution as shown in Eq. 3.
> $O(X):=SXW$
> with the graph input $X\in\mathbb{R}^{n\times d}$, GSO $S\in\mathbb{R}^{n\times n}$, and the parameter matrix $W\in\mathbb{R}^{d\times d}$.
> In this paper, we assume a fully-connected supra-graph on MTS inputs and expect to learn node embeddings and the latent graph structure for capturing the spatial-temporal dependencies in MTS inputs. Accordingly, we do not aim to address a graph shift issue on graph size, node features, or graph structure.
>
> 2) Edge-varying corresponds to the ever-changing correlations of time-series variables over time, i.e., accounting for the dynamic nature of variable dependencies.
> Note that we formulate the MTS forecasting as learning the spatial-temporal dependencies simultaneously on a supra-graph. For input $X\in\mathbb{R}^{N\times T}$, the supra-graph contains $N*T$ nodes of which each node represents the value of each variable at each timestamp in $X$. Such treatment capably captures spatial-temporal dependencies simultaneously and models the time lag effect. The time lag effect between time-series variables is a common phenomenon in real-world MTS scenarios, for example, the time lag influence between two financial assets (e.g., dollar and gold) of a portfolio. It is beneficial but challenging to consider dependencies between different variables under different timestamps [Gilpin2021, Alanqary2021].
> To effectively capture such ever-changing correlations (of time-series variables) over time embodied in the supra-graph, we adopt the edge-varying filters over the supra-graph. The filters adopt different weights to weight the information of different neighbors in each diffusion order and have been demonstrated to be effective to capture the time-varying counterparts [Ifufi2021]. Accordingly, this forms the first attempt to learn the supra-graph to account for the dynamic nature of variable dependencies.
>
> [Gilpin2021] Deep reconstruction of strange attractors from time series. NeurIPS, 2021.
>
> [Alanqary2021] Change Point Detection via Multivariate Singular Spectrum Analysis. NeurIPS, 2021.
>
> [Ifufi2021] Edge varying graph neural networks. TPAMI, 2021.
>
>  3) How does the spectral GCN address the corresponding shifted problem? Are there some motivational examples or intuitions here?
> As we explained above, our proposed EV-FGN is distinct from the spectral-based GCNs and belongs to the spatial-based GCNs. This paper introduces the graph shift operator to formulate a general form of spatial-based graph convolution and subsequently build the edge-varying Fourier graph network based on the edge-varying filters (the edge-varying filters are polynomials of the graph shift operator). Therefore, this paper does not target and address the graph shifted problem.
> We try to figure out the concerns in your comments and try our best to address the concerns. To the best of my knowledge, few papers discussed/studied the graph shifted problem, and we failed to answer the questions. If there is any misunderstanding or you have other concerns, we are always open to further discussion.
>
> We will try our best to further polish our paper to make it better qualify this highly selected community. Thank you again for your time.

---

### Official Review · Reviewer_zvHq · 2022-10-30

**Confidence:** 4
**Correctness:** 3
**Technical Novelty And Significance:** 2
**Empirical Novelty And Significance:** 2
**Recommendation:** 6

**Clarity, Quality, Novelty And Reproducibility:**

The details of the proposed FGSOs need more description. The originality of the work is novel, although limited.

**Strength And Weaknesses:**

## Strength

1. The paper is well-written. I liked how the authors posed the idea of FNO as a simple add-on of GSO to MTS.
2. A decent collection of datasets and experiments were performed, and an extensive set of graph neural net based and transformer based forecasting methods were compared.
3. Additional useful ablation study, hyperparameter sensitivity study, and visualization result analyses were also provided. This provides more interesting information and an understanding of the proposed approaches.

## Weakness:

1. Novelty is somewhat limited. As pointed out there are several works learning graph structure as part of forecasting, this work is a straightforward extension of the original GSO by replacing the original operations with FNO. From the novelty point of view, FNO is a recently proposed method for solving PDEs, and Fedformer (ICML 2022, authors already cited and compared) extends FNO to long-term forecasting. I do not recall any work that combines FNO and graph structure to perform multivariate time series forecasting, but the combination of GSO and FNO in this work is simple.

2. Overall, I found the article tricky to follow. I don’t find the plots in Figure 1 very insightful and it is quite hard to see any informative insights regarding EV-FGN in my opinion. The caption is just "The network architecture of our proposed model". The detailed FGSO is missing in Figure 1, which makes the reader confused about how FNO is integrated with GSO. Based on the codes provided by the authors, I don't see any difference between FGSO and FNO except for the real and imaginary concatenation parts. It seems to me the concatenation operation is different from the proposed EV-FGN layer in equation 12. Please clarify this point. I feel that more need to be said about architecture and how it numerically affects the type of relations that it is capable of representing, i.e., GSO with and without FNO. Also,

3. The high efficiency of FGSO is not obvious. First, I agree the space-invariant property of FGSO, but the claim of scale-free parameters in the Sec 3 remark is unfair. The complexity of FNO is largely determined by the grid size because it is originally proposed to solve PDEs. Instead, the graph size of MTS considered in this paper is fixed. Directly comparing a method solving PDEs and another solving MTS is unfair. Also, it seems to me the complexity of FNO is not $nd^2$. Numerically, based on table 3, GraphWaveNet trained on traffic dataset has a parameter size of 280860 and takes 105.38 seconds per epoch. On the other hand, EV-FGN has a much smaller parameter size but takes a relatively similar amount of time per epoch (99.25 sec). Could you explain why this happen？ I believe the main computation overhead comes from the Fourier transform and its inverse.

## Questions
1. Please use notation consistently throughout the paper. For example, in Sec 3.1, it says 'Given input data $X \in R^{N \times T}$', '2D discrete Fourier transform (DFT) is applied on the embedding of $X \in R^{N \times T \times d}$', but in definition 2, the input size of  $X$ becomes to ${n \times d}$. Why is lowercase $n$ in definition 2 and why is not  ${n \times t \times d}$?
2.  The claim that  Fourier Graph Shift Operator captures high-resolution spatial-temporal dependencies is inaccurate or unclear. The Fourier layer in the Fourier Neural operator on its own only uses low-frequency representations, and thus loses higher-frequency modes. The biased term $W$ might help to recover the higher Fourier modes to some extent, but it can not capture high freq features. This is saying adopting FNO is not to capture high-resolution features. If authors regard 'any two variables at any two timestamps' as high-resolution, I would suggest clarifying high-res in the introduction.




**Summary Of The Paper:**

The paper proposes using Fourier Neural Network (FNO) to combine Graph Shift Operator (GSO), to enable multivariate forecasting.
The architecture first obtains a Fourier transform of embedding of input and then applies it to the proposed operator FGSO, which mainly is consisted of FNO. The authors compare the proposed approach with other Graph based and transformer based forecasting approaches (including ablated versions of the proposed approach) on 7 datasets. They demonstrate competitive performance of the proposed approach.

**Summary Of The Review:**

Overall I think the idea is interesting and in particular, the direction of making multivariate forecasting using FNO is interesting even if it's a simple extension. However, it could be useful to point out why FNO is useful and compatible with GSO and contrast it to related alternate approaches for multivariate forecasting, such as Fedformer. There are several concerns about the method and questions that I feel need to be addressed, i.e., specifics of the method so it could be understood by the readers. Without additional clarifications, I am leaning on the side of rejection. I will wait for the author to clarify before making my final decision.


===============================

After rebuttal:
The authors made additional clarifications and addressed some concerns. I raised my score from 5 to 6

---

> ### Author Response · Authors · 2022-11-09
> **Response to Reviewer zvHq (1/3)**
>
> Dear Reviewer zvHq,
>
> thanks for your valuable comments and we will carefully polish our manuscript according to your comments to make it better qualify the high-quality community. We elaborately address your concerns as follows.
>
> ### W1. About novelty
> **We briefly summarize the novelty of this work as follows**
> + This work forms the first attempt to learn the supra-graph (connecting any two variables at any two timestamps) to account for the dynamic nature of spatial-temporal dependencies.
> + To efficiently compute graph convolutions over the supra-graph, we reformulate the graph convolutions in the Fourier space by leveraging FGSO. To the best of our knowledge, this work makes the first step to reformulate the graph convolutions in the Fourier space.
> + This study makes the first attempt to design a complex-valued feed-forward network in the Fourier space for efficiently computing multi-layer graph convolutions.
>
> **Relation between GSO, FGSO and FNO**
> In our work, we introduce GSO to formulate a general form of the (spatial-based) graph convolution and introduce edge-varying filters (polynomials of the graph shift operator, i.e., GSO) to formulate the multi-layer graph convolutions. Inspired by FNO, we reformulate the graph convolution in the Fourier space with the discrete Fourier transform and define a new Fourier graph shift operator (FGSO). Accordingly, we extend the reformulation to the multi-layer graph convolutions (Proposition 1) and elaborately design a complex-valued feed-forward network in the Fourier space to efficiently compute the multi-layer graph convolutions. This work sheds light on efficiently calculating graph operations in Fourier space by learning Fourier graph shift operators.
> In addition, we adopt frequency-invariant FGSOs corresponding to the (spatial-temporal) space-invariant kernel $\kappa$ reformulate in the design of EV-FGN (a complex-valued feed-forward network) (refer Eqs. 10~12). Compared with FNOs ($\mathbb{C}^{h\times w\times d\times d}$), frequency-invariant FGSOs ($\mathbb{C}^{d\times d}$) not only greatly reduce parameter volumes and save computation costs, and also are empirically proved effective to improve the model generalization of EV-FGN. This leads to the different network architectures of FNO and our EV-FGN: FNO consists of a stack of Fourier layers in which each Fourier layer serially performs 1) DFT to obtain the spectrum of the input, 2) then the graph convolution in the Fourier space, 3) and finally IDFT to transform the output to the time domain (see Fig. 5 in Appendix D.2). Accordingly, FNO needs $K$ pairs of DFT and IDFT for $K$ Fourier layers. In contrast, our EV-FGN needs just a pair of DFT and IDFT at the beginning and end and performs multi-layer graph convolutions on one fly via stacking multiple FGSOs in Fourier space (as shown in Fig. 1). More details can be seen in Appendix D.2.
> In addition, FGSO is also distinct from FNO. Please refer to our response on implementation differences between FNO and FGSO below.
>
> **How we define FGSO from GSO**
> Given a graph G attributed to $X\in \mathbb{R}^{n\times d}$, the graph convolution is defined as $O(X):=SXW$ where $S\in \mathbb{R}^{n \times n}$ is the GSO and $W\in \mathbb{R}^{d\times d}$ is the weight matrix. Regarding the GSO $S$ as $n\times n$ scores, we define a matrix-valued kernel $\kappa: [n]\times [n] \xrightarrow{} \mathbb{R}^{d\times d}$ with $\kappa[i,j]=S_{ij}\circ {W}$, where $[n]=\{1,2,\cdots,n\}$. Then the graph convolution can be viewed as a kernel summation: $O({X})[i]=\sum_{j=1}^n {X}[j]\kappa[i,j] ,\forall i \in [n].$ In the special case of the Green's kernel $\kappa[i,j]=\kappa[i-j]$, the kernel summation can be rewritten as $O({X})[i]=\sum_{j=1}^n {X}[j]\kappa[i-j],\forall i \in [n]$. According to the convolution theorem (the Fourier transform of a convolution of two signals equals the pointwise product of their Fourier transforms), then the graph convolution can be rewritten as $O(X)(i) = \mathcal{F}^{-1}\left(\mathcal{F}(X)\mathcal{F}(\kappa)\right)(i),\forall i \in [n].$ where $\mathcal{F}$ and $\mathcal{F}^{-1}$ are the Fourier transform and its inverse, respectively, and $\mathcal{F}(\kappa)$ is the FGSO and $\kappa[i,j]=S_{ij}\circ{W}$ where $S$ is GSO.

---

> > ### Comment · Reviewer_zvHq · 2022-12-01
> > **Response received**
> >
> > Thank you for your detailed response. It has been received. It addressed some of my concerns and uncertainties.

---

> > > ### Author Response · Authors · 2022-12-02
> > > **Thanks for follow-up feedback**
> > >
> > > Dear Review zvHq,
> > >
> > > Thanks for your valuable time on our manuscript again. If you have any further questions or concerns, please feel free to let us know.
> > >
> > > Authors of Paper 1072

---

> ### Author Response · Authors · 2022-11-09
> **Response to Reviewer zvHq (2/3)**
>
> ### W2. About the readability
> **About Fig. 1**
> Many thanks for your constructive comments. I will provide clear descriptions in the caption in the revised version.
> Here, we briefly explain the network architecture of our EV-FGN according to Fig. 1 and Section 3.1. Given an input $X\in \mathbb{R}^{N \times T}$,
> + 1) Input Embedding: embed $X$ into $\mathbf{X}\in \mathbb{R}^{N\times T \times d}$;
> + 2) 2D FFT: transform $\mathbf{X}$ to Fourier space $\mathcal{X} \in \mathbb{C}^{N \times T \times d}$ via performing 2D DFT on over each discrete $N\times T$ spatial-temporal plane of $\mathbf{X}$;
> + 3) EV-FGN: perform $K$-layer Edge-Varying Fourier Graph Networks (EV-FGN) to achieve multi-layer graph convolutions;
> + 4) 2D IFFT: perform 2D IFFT on the output of EV-FGN over the according spatial-temporal plane;
> + 5) FFN: adopt FFN to generate the multi-step forecasting.
>
> **Implementation differences between FNO and (frequency-invariant) FGSO**
>
> To make it clearer, we provide the codes of FNO2d and FGSO below.
>
> There are three main differences between FNO and FGSO in implementations:
> + As we mentioned above, we adopt frequency-invariant FGSO ($\mathbb{C}^{ d\times d}$) in the EV-FGN, which is different from FNO $\mathbb{C}^{h\times w\times d\times d}$. Therefore, FNO layers and FGSOs are parameterized with complex-valued matrices with different dimensions respectively.
> + After reviewing the codes of FNO, we find that FNO and our FGSO adopt different torch APIs for FFT computations and different approaches to calculate the complex-valued multiplications. Specifically, FNO adopts rfft2 while our FGSO adopts rfft. FNO performs the complex-valued multiplication direct, while our FGSO calculates the real part and imaginary part respectively and then stack (not concatenate) the real part and imaginary part as a complex vector.
> + FNO involves a filter operation to filtrate the higher modes, while FGSO has no such filter operations. As shown in lines 3-4 of the FNO2d code, line 3 only handles the information of [0:self.modes1], and line 4 only handles the information of [-self.modes1:end]. In other words, [self.modes1:-self.modes1] are filtered to zeros. In our code, we do not have such filter operations.
>
> FGSO:
>
> ```python
>    1. x = torch.fft.rfft(x, dim=2, norm='ortho')
>    2. o1_real = torch.zeros(x.shape, device=x.device)
>    3. o1_imag = torch.zeros(x.shape, device=x.device)
>    4. o1_real[:, :, :kept_modes] = (
>         torch.einsum('...bi,bio->...bo', x[:, :, :kept_modes].real, self.w1[0]) - \
>         torch.einsum('...bi,bio->...bo', x[:, :, :kept_modes].imag, self.w1[1]) + \
>         self.b1[0]
>     )
>    5. o1_imag[:, :, :kept_modes] = (
>          torch.einsum('...bi,bio->...bo', x[:, :, :kept_modes].imag, self.w1[0]) + \
>          torch.einsum('...bi,bio->...bo', x[:, :, :kept_modes].real, self.w1[1]) + \
>          self.b1[1]
>     )
>    6. y = torch.stack([o1_real, o1_imag], dim=-1)
>    7. x = torch.view_as_complex(y)
>    8. x = torch.fft.irfft(x, n=L, dim=2, norm="ortho")
> ```
>
> FNO2d:
> ```python
>    1. x_ft = torch.fft.rfft2(x)
>    2. out_ft = torch.zeros(batchsize, self.out_channels,  x.size(-2), x.size(-1)//2 + 1, dtype=torch.cfloat, device=x.device)
>    3. out_ft[:, :, :self.modes1, :self.modes2] = \
>         self.compl_mul2d(x_ft[:, :, :self.modes1, :self.modes2], self.weights1)
>    4. out_ft[:, :, -self.modes1:, :self.modes2] = \
>         self.compl_mul2d(x_ft[:, :, -self.modes1:, :self.modes2], self.weights2)
>    5. x = torch.fft.irfft2(out_ft, s=(x.size(-2), x.size(-1)))
>
>    def compl_mul2d(self, input, weights):
>         return torch.einsum("bixy,ioxy->boxy", input, weights)
> ```
>
> **About Eq. 12**
> Our code is not a concatenation operation of the real and imaginary part, we only stack the real part and imaginary part (as shown in line 6 of our code) as a complex vector that can be transformed in the time domain by ifft (as shown in lines 7-8 of our code).
>
> Note that the calculation of $\mathcal{X}_{k-1}\mathcal{S}_k+b_k$ in Eq. 12 is a complex-valued multiplication. A tiny example: $(a+bi)(c+di)=(ac-bd)+(ad+bc)i$ where $a$ and $c$ are real parts, $b$ and $d$ are imaginary parts, and the imaginary base $i^2=-1$. To achieve the multiplication, we calculate the real part and imaginary part respectively, and then stack the real part and imaginary part as a complex vector. As shown in lines 4-5 of our code, we calculate the real part (i.e., ac-bd) and imaginary part (i.e., ad+bc), respectively. Finally, we stack the real part and imaginary part as a complex vector (i.e., [(ac-bd)+(ad+bc)i]) as shown in lines 6-7.

---

> ### Author Response · Authors · 2022-11-09
> **Response to Reviewer zvHq (3/3)**
>
> ### W3. About efficiency analysis
> Note that we construct a supra-graph with $N\*T$ nodes and a graph size of $(N\*T)\times(N\*T)$ while other GNN-based baselines are associated with an $N$-nodes graph with a graph size of $N\times N$. Despite the tremendous supra-graph, EV-FGN has a lower parameter volume and achieves higher efficiency over the baselines in the experiments. When this is taken into account, the model complexity improvement of EV-FGN in terms of the parameter volume and training time is impressive and desirable.
>
> **Comparison of parameter volumes between FNO and FGSO**
> Note that FGSO and FNO have the same dimension, i.e., $\mathbb{C}^{n\times d\times d}$. But we adopt frequency-invariant FGSO that is scale-free, i.e., $\mathbb{C}^{d\times d}$ being agnostic to the size of MTS inputs. The reason why we adopt frequency-invariant FGSO is not attributed to the size of MTS inputs but the fact that in addition to reducing parameter volumes and saving computation costs, the frequency-invariant parameterized FGSO is empirically proved effective to improve model generalization.
> We argue that the parameter volume of FNO is $O(nd^2)$ when we directly adopt FNO to calculate the graph convolution. This is rational and fair since FNO can replace our defined FGSO to perform graph convolution. In addition, since we construct a supra-graph with $N*T$ nodes, we can have different graph sizes corresponding to different window-sized MTS inputs, i.e., different $T$. Besides, when we build and train the networks in PDE and MTS problems, both the grid size in PDEs and the graph size in MTS are fixed. To this point, the grid size is also fixed. In our opinion, the comparison of parameter volumes between FNO and FGSO is fair. Considering the high volume of parameters of FNO, especially in the case of multi-layer graph convolutions, we define FGSO and adopt the frequency-invariant FGSO to design our EV-FGN for efficient multi-layer graph convolutions.
>
> **About parameter volume and efficiency**
> The low parameter volume of our EV-FGN is largely attributed that we adopt frequency-invariant FGSOs that are scale-free, i.e., $\mathbb{C}^{d\times d}$ being agnostic to the size of MTS inputs. The main computation of EV-FGN comes from the 2D FFT and 2D IFFT and the subsequent FFN. Note that our model consists of three parts, the embedding layer, the EV-FGN, and the FFN.
> Reasonably, a large volume of parameters must not lead to low training efficiency. For example, the parameters of AGCRN are three times larger than GraphWaveNet, while its training time is similar to that of GraphWaveNet, and the Transformer-based baselines have much larger parameters than GNN-based baselines, but they have less training time, shown as below.
> |Models | parameters| epoch time(s)|Models | parameters| epoch time(s)|
> |:----:|:----:|:----:|:----:|:----:|:----:|
> |FEDformer(2022)| 16211906 | 17.68$\pm$0.49 |AGCRN(2020)|749940|113.46$\pm$1.91|
> |Autoformer(2021)|15425474 | 13.10$\pm$0.71 |StemGNN(2020)|1606140|185.86$\pm$2.22|
> |Informer(2021)|14750658 | 11.76$\pm$0.52 |GraphWaveNet(2019)|280860|105.38$\pm$1.24|
> | | | |Our model |190564|99.25$\pm$1.07|
>
> ### Responses to the questions
> **Q1**
> In Sections 3.2 and 3.3, we define the graph convolution, FGSO, and formulate Proposition 1 in a general scenario with input $X\in\mathbb{R}^{n\times d}$, which is more convenient and easier to understand. When applying to the 2D space, i.e., the spatial-temporal space of multivariate time series, we have $n=N*T$ and the discrete space $[n]=[N]\times [T]$ as we discussed in Section 3.2 (below Definition 2) and Section 3.3 (in Remarks).
> Thank you for pointing out this. We will supplement the explanation at the beginning of Section 3.2 and Section 3.3 to avoid confusion.
>
> **Q2**
> Yes, we regard “any two variables at any two timestamps” as high-resolution. Note that we formulate the MTS forecasting as learning the spatial-temporal dependencies on a supra-graph connecting any two variables at any two timestamps. The high-resolution relates to the granularity of spatial-temporal dependencies. Compared to previous MTS models, e.g., GNN-based models focusing on spatial dependencies (variable-wise) and Transformer-based models focusing on temporal dependencies (timestamp-wise), our EV-FGN model has fine-grained spatial-temporal dependencies via attending to any two variables at any two timestamps (value-wise). Thank you for your suggestion. We will clarify the term ‘high-resolution’ in the introduction for a better understanding. Moreover, FNO only uses the lower Fourier modes and filters out the higher modes of the input by a linear transform. However, our model has not any filter operations to the input, and the full spectrum of the input are fed into our model. More details can be seen in the previous implementation comparison part of our response
>
> Thank you again for your time and precious advice.

---

### Official Review · Reviewer_iyiB · 2022-11-01

**Confidence:** 3
**Correctness:** 4
**Technical Novelty And Significance:** 3
**Empirical Novelty And Significance:** 3
**Recommendation:** 6

**Clarity, Quality, Novelty And Reproducibility:**

As described above, the proposed method has novelty, however the presentation, the motivation and the clarity in some concepts can be improved. The paper and the appendix of the paper have a lot of extra results and analyses, as well as, implementation details and links to the publicly available code repositories of the comparison models and the datasets, that will help with the reproducibility. They have also submitted their code in the supplementary material.

**Details Of Ethics Concerns:**

No ethical concerns.

**Strength And Weaknesses:**

Strengths:
- The paper is about an interesting topic and the authors do a good job to describe the problem and their methodology, as well as their contributions.
- There is novelty in this work. The authors build the proposed model in existing concepts, however this is the first attempt to design a complex-valued feed-forward network in the Fourier space with a focus on reducing the complexity that the use of a supra-graph is introducing.
- The structure of the paper is well-defined. The paper has a nice flow. The appendix also has interesting information and more details, that are useful when reading the paper.
- The authors have done an extended evaluation of the proposed model and 13 comparison state of the art models in 7 datasets from different applications.
- The analysis in 4.3 is nicely done and highlights the improvements and the need of the FGSO to reduce the complexity and the runtime of the proposed methodology.

Weaknesses:
- This work might have a limited interest to researchers of ICLR as the main focus is to efficiently calculate graph operations in the Fourier space.
- The paper can be difficult to follow. The concepts presented in this work are hard to understand, especially if the reader does not have previous familiarity with them. The authors can do a better job in introducing these concepts with a few examples of how they are used in the literature. Specifically, more motivation in the introduction section is needed and specific examples of the challenges that the papers/methods that are cited in the related work section face, and how the proposed methodology will avoid such challenges. Also, a few examples in the introduction of each concept in the methodology section will help the reader understand and remember the notation and the need of each part/concept.
- It would be interesting to see an evaluation for the multivariate time series forecasting using state of the art methods in time series representation learning as well.
- It would also be interesting to compare the proposed methodology when not using the FGSO, how does the performance change and how does the complexity/runtime/number of parameters change?

- Minor typos:
— The word weight when used as a verb is noted as “weigh” at least two times.
— The dataset METR-LA is noted as META-LA in Table 1.

**Summary Of The Paper:**

This paper is about the combination of relations of the variables in multivariate time series (MTS) and graph neural networks for the analysis and prediction of MTS. The proposed method is to build a fully-connected supra-graph that connects variable at any two timestamps to learn high-resolution variable dependencies in an efficient way using graph neural networks. One of the contributions is the shift operator that is being proposed to reduce the computation complexity that is introduced by the supra-graph. Specifically, the authors propose to construct the Edge-Varying Fourier Graph Networks (EV-FGN) with Fourier Graph Shift Operator (FGSO) to perform graph convolutions in the frequency domain on a lower-complexity. The experimental setup includes seven datasets from different application scenarios and thirteen comparison models. The overall evaluation shows that in most cases the proposed model EV-FGN outperforms the other state of the art models. The authors include an analysis on the number of parameters and the training time, to show how the FGSO achieves an improvement in the complexity of the model. The authors also include a visualization analysis to show the patio-temporal representation that is learnt by the EV-FGN.

**Summary Of The Review:**

Overall, this paper introduces a methodology that for the multivariate timeseries analysis and forecasting, using the supra graph and Fourier Graph Shift Operator to reduce the complexity of such a graph. The proposed method outperforms 13 other state of the art models when comparing to 7 datasets of various domains. The introduced methodology and concepts are hard to follow, but the authors try to introduce them, describe them and justify their decisions in a satisfactory way. However, improvements can be made to help the readers that are not familiar with these concepts (see the weaknesses section above for suggestions.).

---

> ### Author Response · Authors · 2022-11-09
> **Response to Reviewer iyiB**
>
> Dear Reviewer iyiB,
>
> Many thanks for your constructive comments and positive feedback. We will address your concern as below and carefully polish the manuscript to make it better qualify this highly selected community.
>
> **1. About the limited interest to the researchers of ICLR**
>
> Graphs naturally appear in numerous application domains, ranging from social analysis, and bioinformatics to computer vision, and graph convolutional networks have a great expressive power to learn the graph representations and have achieved superior performance in a wide range of tasks and applications, e.g., recommender systems, link prediction, MTS forecasting, message passing et al. [Phan2022,Wu2022,Yuan2022]. This work proposes an efficient approach to calculate graph operations in the Fourier space, which might be interesting to the graph learning community. In addition, we build a novel network EV-FGN for MTS forecasting that achieves SOTA performance and high efficiency. This work provides a new learning paradigm and a SOTA benchmark for the MTS community. Graph learning and time-series representation learning are in the scope of ICLR and are also popular in ICLR. In light of this, this work might be interesting to the researchers of ICLR, to my knowledge.
>
> [Phan2022] Aspect-level sentiment analysis: A survey of graph convolutional network methods, Information Fusion, 2022
> [Wu2022] Graph Neural Networks in Recommender Systems: A Survey, ACM Computing Surveys, 2022.
> [Yuan2022] Explainability in graph neural networks: A taxonomic survey, TPAMI, 2022.
>
> **2. About Evaluation**
>
> In the evaluation, there are mainstream and related approaches, i.e., transformer-based and GNN-based models. We compared our model with SOTA GNN- and Transformer-based models to demonstrate the superiority of our proposed EV-FGN. In the visualization experiments, we also evaluate the spatial-temporal node embeddings learned by our EV-FGN from different aspects.
> Regarding time series representation learning methods, we will further conduct experiments to compare our model with the SOTA time series representation model and report the results in our revised version.
>
> **3. About paper readability**
>
> Many thanks for your constructive suggestions. We will carefully revise our manuscript, especially the introduction and methodology parts, to deliver a better understanding of our work.
>
> **4. About efficiency without FGSO**
>
> Since we build a supra-graph connecting any two variables at any two timestamps to represent non-static spatial-temporal dependencies, the supra-graph is with $N*T$ nodes, and the complexity of conducting graph convolutions over it will be $O((N\*T)^2)$ if without FGSO. Compared with previous GNN-based models (with $N$ nodes) that have $O(N^2)$ complexity, our model without FGSO will have relatively higher complexity.
>
> Moreover, in Table 3, we compared our model with those GNN-based models which are with $N$ nodes, and the results showed that our model with $N*T$ nodes also has fewer parameters and lower training time. This demonstrates that our model with FGSO will be much more efficient than our model without FGSO.
>
> **5. About typos**
>
> Thank you for pointing out the typos. We will thoroughly proofread the manuscript and carefully correct the typos in the revised version.
>
> Thank you again for your time and precious advice. We will polish our manuscript to make it better qualify this highly selected communityand upload the revised version as soon as possible.

---

> > ### Comment · Reviewer_iyiB · 2022-11-23
> > **Thank you**
> >
> > I would to acknowledge that I have read the authors' responses to all reviews. I would to thank the authors for taking the time to respond to us and for reviewing the paper and making changes based on our suggestions.

---

> > > ### Author Response · Authors · 2022-12-02
> > > **Thanks for follow-up feedback**
> > >
> > > Dear Review iyiB,
> > >
> > > Thanks for  your constructive comments and valuable time on our manuscript again. If you have any further questions or concerns, please feel free to let us know.
> > >
> > > Authors of Paper 1072

---

> ### Author Response · Authors · 2022-11-11
> **Additional experimental reports with SOTA time series representation learning model CoST**
>
> Dear iyiB,
>
> Following your suggestions, we perform the SOTA model in the time series representation learning, CoST (ICLR 2022)[1], which separates the representation learning and downstream forecasting task and proposes a contrastive learning framework that learns disentangled season-trend representations for time series forecasting tasks. We compare our model with CoST on the COVID-19 and METR-LA datasets under different prediction lengths (3, 6, 9 and 12) and the results are reported in the following table.
>
> COVID-19
>
> |Metrics | MAE| RMSE|MAPE(%) | MAE| RMSE|MAPE(%)| MAE| RMSE|MAPE(%)| MAE| RMSE|MAPE(%)|
> |:----:|:----:|:----:|:----:|:----:|:----:|:----:|:----:|:----:|:----:|:----:|:----:|:----:|
> |prediction length | |3| | |6| | |9| | |12|
> |GraphWaveNet| 0.092 | 0.129 |53.00 | 0.133| 0.179 | 65.11| 0.171 |0.225 | 80.91| 0.201 |0.255|100.83
> |StemGNN|  0.247 |0.318| 99.98| 0.344 |0.429| 125.81| 0.359 |0.442|131.14 | 0.421 |0.508|141.01
> |AGCRN| 0.130 | 0.172 | 68.64 | 0.171 | 0.218 | 79.29 | 0.224 | 0.277 | 113.42 | 0.254 | 0.309 | 125.43
> |MTGNN| 0.276 |0.379|91.42 | 0.446 |0.513| 133.49| 0.484 |0.548| 139.52| 0.394 |0.488|88.13
> |TAMP-S2GCNets| 0.140 | 0.190 | 50.01 | 0.150 | 0.200 | 55.72| 0.170 | 0.230 | 71.78 | 0.180 | 0.230 | 65.76
> |CoST| 0.122 | 0.246 | 68.74 | 0.157 | 0.318 | 72.84 | 0.183 | 0.364 | 77.04 | 0.202 | 0.377 | 80.81
> |EV-FGN(ours)| 0.071 | 0.103 | 61.02 | 0.093 | 0.131 | 65.72 | 0.109 | 0.148 | 69.59 | 0.124 | 0.164 | 72.57
>
> METR-LA
>
> |Metrics | MAE| RMSE|MAPE(%) | MAE| RMSE|MAPE(%)| MAE| RMSE|MAPE(%)| MAE| RMSE|MAPE(%)|
> |:----:|:----:|:----:|:----:|:----:|:----:|:----:|:----:|:----:|:----:|:----:|:----:|:----:|
> |prediction length | |3| | |6| | |9| | |12|
> DCRNN | 0.160 | 0.204 | 80.00 | 0.191 | 0.243 | 83.15 | 0.216 | 0.269 | 85.72 | 0.241 | 0.291 | 88.25
> STGCN | 0.058 | 0.133| 59.02 |0.080 |0.177 |60.67 |0.102 | 0.209| 62.08 |0.128 |0.238 |63.81
> GraphWaveNet | 0.180 |0.366 | 21.90 | 0.184 |0.375 |22.95 | 0.196 | 0.382 | 23.61| 0.202 |0.386 |24.14
> MTGNN | 0.135 | 0.294 | 17.99 | 0.144 |0.307 | 18.82 | 0.149 |0.328 |19.38 | 0.153 | 0.316 |19.92
> StemGNN | 0.052 | 0.115 | 86.39 | 0.069 | 0.141 | 87.71 |0.080 | 0.162 | 89.00| 0.093| 0.175 |90.25
> AGCRN | 0.062 | 0.131  | 24.96 | 0.086 | 0.165 | 27.62 | 0.099 | 0.188 | 29.72 | 0.109 | 0.204 | 31.73
> Informer | 0.076 |0.141 | 69.96 | 0.088 | 0.163 | 70.94 | 0.096 |0.178  |72.26| 0.100 |0.190 |72.54
> CoST |  0.064 |0.118 | 88.44 | 0.077 |0.141| 89.63 |0.088 |0.159| 90.56| 0.097 |0.171 |91.42
> EV-FGN(ours) | 0.050 |0.113| 86.30|0.066|0.140|87.97 |0.076|0.159| 88.99|0.084|0.165| 89.69
>
> From the table, we can find CoST has achieved competitive performances since it can learn disentangled discriminative seasonal and trend representations via contrastive learning. However, our model still outperforms CoST. Because, compared with CoST, our model not only can learn the temporal representations, but also can capture the discriminative spatial representations. Fig. 2 in our paper shows our model can learn highly interpretative spatial correlations, and Figs. 3 and 10 in our paper demonstrate that our model can exploit the time-varying and discriminative temporal dependencies.
>
> We will add this experimental results in our revised submission. Thanks for your kind suggestions.
>
> [1] CoST: Contrastive Learning of Disentangled Seasonal-Trend Representations for Time Series Forecasting. ICLR 2022
> Gerald Woo, Chenghao Liu, Doyen Sahoo, Akshat Kumar, Steven C. H. Hoi

---

### Official Review · Reviewer_jKjm · 2022-11-01

**Confidence:** 3
**Correctness:** 3
**Technical Novelty And Significance:** 3
**Empirical Novelty And Significance:** 2
**Recommendation:** 5

**Clarity, Quality, Novelty And Reproducibility:**

The proposed idea is clearly novel and well-motivated. The writing is mostly clear, however, there are still some questions regarding the clarity (and non-triviality) of the proposed formalism - see above. I have not tried running the experiments, however, out of the reported 6 datasets, dataloaders for only 3 of them are provided in the supplementary material. This excludes complete reproducibility of the results.

**Strength And Weaknesses:**

### The paper has several strengths:
- Efficient GNN parametrization in the Fourier domain resulting in reduced computational complexity compared to naive graph-based approaches for MTS modelling.
- Consequently, the proposed algorithm allows for joint modelling of spatial and temporal variables, that would be prohibitive otherwise due to computational restrictions.
- Experiments show SOTA performance on a variety of datasets with better computation time and reduced parameter count.
- Ablation studies demonstrate the effectiveness of each proposed model component.

### It also has some weaknesses in my opinion:
- The clarity of the formulation is often obfuscated by confusing terminology.

For example, at the end of page 4: "Accordingly, we can parametrize FGSO with a complex-valued matrix which is space-invariant..."
This sentence seems misleading. The same term is used previously to refer to a regular GSO, in which case it meant that the kernel function $\kappa$ is shared across nodes $[n]$. However, the invariance in this case is with respect to the frequency input in the Fourier domain. Therefore, it is not space but frequency-invariant. It seems to me that this is actually the point that allows the convolution operator to be non-static in time since frequency-invariance in the Fourier domain still allows the convolution to be temporally varying in the input domain. This point could be emphasized more.

Similarly, I was confused by the extension of the argument by considering a 2D domain (i.e. from $[n]$ to $[N] \times [T]$). The authors say that "we can extend Definition 2 to a 2D discrete space". How would that look like? In that case, would the proposed parametrization invariant be invariant to both frequency components?

In Proposition 1, the dimensions associated with the $S_k$ and $\mathcal{S}_k$ operators seem to be confusing. $S_k$ is defined to be $n \times n$, while $\mathcal{S}_k$ is defined as $n \times d \times d$. What is the relation between the two?

On a related note, the left-hand side of equation (9) should probably read as $H_{EV} X W$ since (I assume) that it is the edge-varying counter part of equation (7).


- No interpretation is provided for the frequency-invariant parametrization

 Related, I was somehow lacking an interpretation of what this shared parametrization across frequencies means with respect to the convolution operation in the input domain. Although it allows for a non-static temporal representation, it clearly places a big restriction on the possible spatio-temporal interactions that it can represent. I guess my worry is that it might reduce to something straightforward in the input domain, which is only obfuscated by the dual formalism. What do we get when we invert this? What's the range of operations that it can represent? These are questions not at all addressed at the moment.

- Experimental improvements are often marginal

In the experiments, the improvement is often marginal over the baselines. Further, in Appendix E.4 the authors report that they carefully fine-tune the hyperparameters of their own model. However, for the baselines in Appendix E.2, they report the use recommended default settings only. Hence, significantly more effort is spent on tuning their model compared to zero tuning on the baselines, which makes the experimental results inconclusive. I would be more convinced if rather than using default settings, they also fine-tuned the baseline settings at least with respect to the hyperparameters deemed significant by the respective authors of these baselines.

**Summary Of The Paper:**

The paper proposes a method for discovering the underlying correlations between variables in a temporal system which drive its evolution. Recent works have applied GNNs to model these relationships, but several of these approaches have employed a static graph over time failing to capture the potential evolution of this dependency structure. To address this, the proposed approach utilizes an all encompassing supra-graph that jointly represents relationships between all variables at all timesteps. Towards this, the notion of Fourier Graph Shift Operator is introduced which performs graph convolution in the frequency domain. This representation is exploited in devising a frequency-invariant convolution operation that can avoid the $n^2$ node complexity associated with a dense graph and effectively reduces
 it to $n \cdot \log n$. Edge-varying diffusion networks are introduced for modelling higher-order diffusion over multiple neighbourhood sizes and a corresponding Fourier representation is introduced.

**Summary Of The Review:**

The paper proposes a novel approach for performing graph convolution in the Fourier space at a reduced computational cost. This speed-up allows for joint modelling of spatio-temporal interactions. The flexibility of the introduced parametrization, however, is not investigated, and consequently there is a lack of theoretical understanding of what the approach can actually learn. Experiments show minor improvements over the considered baselines, but I think overall these are not conclusive under the given experimental setting.

---

> ### Author Response · Authors · 2022-11-09
> **Response to Reviewer jKjm (1/3)**
>
> Dear Reviewer jKjm,
>
> Many thanks for your constructive comments. We will carefully address your concerns as below.
>
> **1. Indeed, the kernel $\kappa$ is spatial-temporal space-invariant, and FGSO is frequency-invariant.**
>
> Thanks for pointing out the misleading expression. FGSO derived from the space-invariant is frequency-invariant in the Fourier space. As you mentioned, despite of the frequency-invariant FGSO, our EV-FGN can still capture spatial-temporal dependencies embodied in the inputs.
> Specifically, we formulate the MTS forecasting as learning the spatial-temporal dependencies simultaneously on a supra-graph. For input $X\in\mathbb{R}^{N\times T}$, we assume the fully-connected supra-graph containing $N*T$ nodes (of which each node represents the value of each variable at each timestamp in $X$) and perform $K$-layer EV-FGN in the frequency domain (for efficiently computing multi-layer graph convolutions over the supra-graph) to learn the latent graph structure. The supra-graph accounts for **the time lag effect and the dynamic nature of variable dependencies over time**, which facilitates capturing the spatial-temporal dependencies in MTS.
>
> **2. Explanation to the extension of Definition 2 to 2D domain**
>
> Recall Eqs. 4~5:
>
> Eq. 4 (kernel summation): $O({X})[i]=\sum_{j=1}^n {X}[j]\kappa[i,j]  \quad\quad   \forall i \in [n]$
>
> Eq. 5 (kernel summation):$O({X})[i]=\sum_{j=1}^n {X}[j]\kappa[i-j]=(X*\kappa)[i]  \quad\quad   \forall i \in [n]$
>
> When we extend the equations to 2D domain, i.e., from $[n]$ to $[N]\times[T]$, it means performing a kernel summation/graph convolution over the discrete spatial-temporal space corresponding to all nodes in the supra-graph. Obviously, these computations of the kernel summation can be easily extended to 2D domain.
> Similarly, according to the convolution theorem, we can obtain a 2D-version of Eq. 6:
> Eq. 6 (graph convolution): $O(X)(i) = \mathcal{F}^{-1}\left(\mathcal{F}(X)\mathcal{F}(\kappa)\right)(i)\quad\quad  \forall i \in [N]\times[T]$
> with $\mathcal{F}$ and $\mathcal{F}^{-1}$ denote the 2D discrete Fourier transform and its inverse, respectively.
> Accordingly, when extending Definition 2 to 2D domain, $\mathcal{F}$ denotes the 2D discrete Fourier transform. Therefore, given input embeddings $\mathbf{X}\in\mathbb{R}^{N\times T\times d}$, we perform 2D discrete Fourier transform on each discrete $N\times T$ spatial-temporal plane of the embeddings to obtain the frequency spectrum, and then feed the frequency input into K-layer EV-FGN followed by two-layer (real-valued) feed-forward networks to generate multi-step forecasting (see Section 3.1 for more details). Note that we adopt the frequency-invariant FGSO ($d\times d$) in the EV-FGN where the feed-forward computations act on the embedding dimension, i.e., $d$.
> In addition, when extending Definition 2 to 2D domain, the frequency-invariant FGSO is invariant to both frequency components derived from the spatial dimension ($N$) and time dimension ($T$) respectively.
>
> **3. About the dimensions of $S_k$ and $\mathcal{S}_k$ in Proposition 1.**
>
> Referring to Definition 2, $S_k\in\mathbb{R}^{n\times n}$ is the $k$-th **GSO**, and $\mathcal{S}_k\in\mathbb{C}^{n\times d\times d}$ is the $k$-th **FGSO** corresponding to $S_k$. It satisfies $\mathcal{F}(S\_k {X})=\mathcal{F}({X})\times\_n\mathcal{S}\_k$.
> Recall the definition of the matrix-valued kernel $\kappa: [n]\times [n] \xrightarrow{} \mathbb{R}^{d\times d}$ with $\kappa[i,j]=S\_{ij}\circ {W}$, where $[n]=\{1,2,\cdots,n\}$. In Definition 2, We define $\mathcal{S}:=\mathcal{F}(\kappa)$ as a Fourier graph shift operator (FGSO). Intuitively, we can parameterize each $\mathcal{S}$ with a complex-valued matrix with dimensions $\mathbb{C}^{n \times d \times d}$.
>
> **4. About Eq. 9.**
>
> Yes, the left-hand side of Eq. 9 is $H_{EV}XW$ derived from Eq. 7. In Proposition 1, we omit the weight matrix $W$ for convenience, that is, we treat $W$ as an identity matrix. Please refer to Appendix C.1 *Proof of Proposition 1*. This treatment is feasible since the weight matrix $W$ can be absorbed in the embedding input, precisely the embedding matrix parameters. Note that we feed the input embeddings into EV-FGN (refer to Fig. 1 and Section 3.1). In addition, this treatment will not reduce the capability of EV-FGN intuitively since we adopt edge-varying filters in EV-FGN. Note that traditional GCNs adopt different weight matrices (regarding the model capability) but the same GSO (e.g., adjacency and Laplacian matrices regarding the given graph structure) in different diffusion orders. Differently, there is no pre-given graph structure in MTS forecasting scenarios, therefore we adopt the edge-varying filters, i.e., varying GSOs in EV-FGN, which does not reduce the model capability and achieves desirable performance empirically.

---

> ### Author Response · Authors · 2022-11-09
> **Response to Reviewer jKjm (2/3)**
>
> **5. The interpretation of frequency-invariant parameterization.**
>
> In addition to reducing parameter volumes and saving computation costs, the frequency-invariant parameterized FGSO is empirically proved effective to improve model generalization.
> As we mentioned above, we perform EV-FGN over the input embeddings and adopt the frequency-invariant FGSOs. Relatively to directly adopting the frequency-variant FGSOs ($\mathbb{C}^{n\times d\times d}$), we subtly “factorize” the frequency-variant parameterization to the time domain (i.e., the embedding matrix $\Phi\in\mathbb{R}^{n\times d}$) and the frequency domain (i.e., frequency-invariant FGSO $\mathbb{C}^{d\times d}$). In the time domain, we embed the raw MTS inputs to improve the model learning capability, while we learn the same transformation (FGSO) for all $N*T$ frequency points in the frequency domain (similar to CNN with shared-weight convolution kernels or filters that slide along input features). Note that the frequency spectrum in the frequency domain has a global view of which each frequency point attends to all variables or timestamps. This treatment in EV-FGN guarantees the model capacity and is empirically proved superior over the treatment without embeddings and with frequency-variant FGSO (please refer to the **Ablation study** for detailed results). Although the frequency-variant parameterization may be more powerful and flexible than the frequency-invariant one, it introduces more parameters in the frequency domain, especially for multi-layer EV-FGN, and may not obtain superior performance due to inadequate training or overfitting.
>
> *Why does the frequency-invariant FGSO correspond to space-invariant kernel?*
>
> According to Eqs. 3-7, We define FGSO as $\mathcal{S}:=\mathcal{F}(\kappa)$ and $\kappa[i,j]=S_{ij}\circ{W}$ where $S$ is GSO and $W$ is the weight matrix. A space-invariant kernel means that $\kappa[i,j]$ is invariant to $i$ and $j$, that is the kernel is irrelevant to $S_{ij}$, i.e., the GSO. Accordingly, $\kappa[i,j]=s^0\circ{W}$ with an arbitrary constant $s^0$, and we can parameterize $\mathcal{F}(\kappa) \in\mathbb{C}^{d\times d}$ being frequency-invariant.
>
> **6. What do we get when we invert this? What's the range of operations that it can represent?**
>
> When we perform 2D IFFT on the outputs of EV-FGN, we will obtain the node embeddings, i.e., $\mathbf{X}_{\Psi}\in \mathbb{R}^{n\times d}$. The output node embeddings are obtained via multi-layer graph convolutions and involves the neighbor information during the multi-order diffusion.
> Note that both frequency-invariant ($\mathbb{C}^{d\times d}$) and frequency-variant ($\mathbb{C}^{n\times d\times d}$) FGSOs satisfy the convolution theorem. The difference is that they correspond to different convolutional filters in the time domain. Since we introduce the embedding dimension, i.e., $d$, it allows us to design different types of FGSO (corresponding to different kernels $\kappa$), for example frequency-variant $\mathbb{C}^{n\times d\times d}$, $\mathbb{C}^{n\times d}$, and $\mathbb{C}^{n}$, and frequency-invariant $\mathbb{C}^{d\times d}$. In our experiments, we have tried different kinds of FGSO, corresponding to different convolutional filters, and finally adopt the frequency-invariant FGSO that has superior performance and less parameters.
> Thank you for your helpful suggestion. We will add more explanations and insightful discussions in the revised version to deliver a clear understanding of our work.
>
> **7. About experimental settings**
>
> In addition to the default hyperparameters of the benchmarks recommended by their authors in their paper, we further tune their hyperparameters on different datasets according to their recommended parameter search ranges respectively. Since the seven benchmark datasets in the experiments are commonly used in MTS forecasting and adopted by the baseline models, the default settings achieve the best performance in most cases. Some exceptions: for example, we tune and set the hyperparameter settings for DeepGLO and TAMP-S2GCNets on the COVID-19 dataset and adjust GraphWaveNet. In Appendix E.2, we show the default hyperparameter settings along with the settings after our tuning experiments.

---

> ### Author Response · Authors · 2022-11-09
> **Response to Reviewer jKjm (3/3)**
>
> **8. About experimental results.**
>
> In the evaluation, we adopt 13 mainstream and related approaches for comparison, including traditional models and SOTA LSTM-based, CNN-based, GNN-based, and Transformer-based models. Extensive experiments on seven real-world datasets show the superiority of our EV-FGN over all the SOTA baselines. Specifically, EV-FGN achieves 2.4%\~23.6% MAE improvement and 2.5%\~24.8 RMSE improvement over the best baselines. The SOTA performance EV-FGN is impressive and desirable, especially considering the high efficiency of EV-FGN over the SOTA GNN-based models.
>
> **9. About reproducibility.**
>
> There are two dataloaders (named Dataset_ECG and Dataset_Dhfm) in the source code (precisely in the data_loader.py). **Note that we adopt the dataloader Dataset_Dhfm for wiki, Solar, and traffic and adopt the dataloader Dataset_ECG for the other datasets**. In addition, we only upload three datasets (i.e., traffic, ECG, and Covid) due to the limitation of maximum file size (100M), while other datasets are too large to upload. Thanks for your constructive comments. We will add detailed annotations in our source codes for better understanding and reproducibility and upload a new copy of the source codes.
>
> Thank you again for your time and precious comments.We will carefully polish the manuscript according to your comments.

---

### Official Review · Reviewer_VMXz · 2022-11-03

**Confidence:** 3
**Correctness:** 3
**Technical Novelty And Significance:** 3
**Empirical Novelty And Significance:** 3
**Recommendation:** 5

**Clarity, Quality, Novelty And Reproducibility:**

Clarity: Good
Quality: Fair
Novelty: Limited
Reproducibility: Good


**Strength And Weaknesses:**

The results are adequate in various datasets and look great. The training details are comprehensive. The equations are proved clearly in the paper.

But The idea seems limited, GSO and MTS with Fourier transform existed before, the paper just extended to other latent space such as Fourier space. The motivation of this is not convincing enough.

Questions
1.	Why the results on MAPE are higher than other baselines on several datasets?
2.	In Table 3, the authors compare parameter efficiency with old baselines, why not compare with recent baselines from 2021 such as FEDformer?
3.	Figure 1 can not visualize the proposed method clearly.


**Summary Of The Paper:**

This paper adaptively learns a supra-graph, representing non-static correlations between any two variables at any two timestamps, to capture high-resolution spatial-temporal dependencies, and define FGSO that has the capacity of scale-free learning parameters in the Fourier space. Accordingly, authors construct a complex-valued feed forward network, dubbed as Edge-Varying Fourier Graph Networks (EV-FGN), stacked with multiple FGSOs to perform high-efficiency multi-layer graph convolutions in the Fourier space.

**Summary Of The Review:**

The empirical results are good, while the analytical results are limited.

---

> ### Author Response · Authors · 2022-11-09
> **Response to Reviewer VMXz (1/2)**
>
> Dear Reviewer VMXz,
>
> Many thanks for your helpful and constructive comments. We will carefully address your concerns and questions in detail.
>
> ### Comments:
> But The idea seems limited, GSO and MTS with Fourier transform existed before, and the paper just extended to other latent space such as Fourier space. The motivation of this is not convincing enough.
>
> ### Response:
>
> **1. The motivation of our proposed EV-FGN is twofold.**
>
> - **The benefits of the supra-graph**. We formulate the MTS forecasting as learning the spatial-temporal dependencies simultaneously on a supra-graph. For input $X\in\mathbb{R}^{N\times T}$, the supra-graph contains $N*T$ nodes of which each node represents the value of each variable at each timestamp in $X$. In addition to the spatial-temporal dependencies, the supra-graph accounts for the time lag effect and the dynamic nature of variable dependencies over time, which intuitively and empirically facilitates multivariate time series forecasting.
>
> - **The motivation of EV-FGN**. Since the supra-graph is tremendous with a graph size of $(N\*T) \times (N\*T)$, it leads to an extremely high time complexity of graph convolution. To **efficiently** capture the ever-changing spatial-temporal dependencies in the supra-graph, we reformulate the general spatial-based graph convolution in the frequency domain via discrete Fourier transform and propose the edge-varying Fourier graph network (EV-FGN).
>
> **2. We summarize the novelty of this work as follows.**
>
> + This work forms the first attempt to learn the supra-graph to account for the dynamic nature of spatial-temporal dependencies.
> + To efficiently compute graph convolutions over the supra-graph, we reformulate the graph convolutions in the Fourier space by leveraging FGSO. To the best of our knowledge, this work makes the first attempt to reformulate the graph convolutions in the Fourier space.
> + This study makes the first attempt to design a complex-valued feed-forward network in the Fourier space for efficiently computing multi-layer graph convolutions.
>
> **3. Compared with previous MTS work with Fourier transform**.
>
> Our work reformulates the graph convolution in the frequency domain according to the Convolution Theorem, precisely we achieve the time-consuming graph convolution (in the time domain) via the efficient element-wise multiplication in the frequency domain. However, the previous MTS works generally adopt Fourier transform to obtain frequency spectrum for feature enhancements or to calculate the time-series correlations. In addition, those methods separately model the spatial and temporal dependencies, while our proposed EV-FGN captures the spatial-temporal dependencies simultaneously. Please refer to Section 2.1 for more details.
>
> *Convolution Theorem: states that the Fourier transform of a convolution of two functions (or signals) equals the point-wise product of their Fourier transforms*.
>
> **4. Compared with previous work on GSO**.
>
> Our work utilizes GSO to formulate the general form of graph convolutions, which subsequently facilitates reformulating the graph convolutions in the frequency domain leveraging discrete Fourier Transform. However, previous work on GSO generally adopts GSO to perform graph shift operations in the time domain (see more details in Section 2.2).
> In summary, although our work introduces the definition of GSO and adopts Fourier transform, it is distinct from previous work and makes the first attempt to efficiently compute multi-layer graph convolutions in the Fourier space.

---

> ### Author Response · Authors · 2022-11-09
> **Response to Reviewer VMXz (2/2)**
>
> ### Questions
>
> 1. Why the results on MAPE are higher than other baselines on several datasets? 2. In Table 3, the authors compare parameter efficiency with old baselines, why not compare with recent baselines from 2021 such as FEDformer? 3. Figure 1 can not visualize the proposed method clearly.
>
> ### Response
>
> **Q1**. Why the results on MAPE are higher than other baselines on several datasets?
>
> **A1**. Note that we formulate the MTS forecasting as learning the spatial-temporal dependencies simultaneously on a supra-graph in which each node represents the value of each variable at each timestamp. In addition, the supra-graph accounts for the time lag effect and the dynamic nature of variable dependencies over time, which are beneficial and challenging for multivariate time series forecasting.
> Accordingly, compared with the transformer-based models, our model considers the spatial dependencies explicitly in addition to the temporal dependencies. Compared with the GNN-based models, our model learns comprehensive spatial-temporal dependencies simultaneously and attends to time-varying dependencies among variables.
> The visualization part in 4.4 also demonstrates that our model can learn discriminative spatial and temporal dependencies and exploit the time-varying dependencies among variables.
>
> **Q2**. In Table 3, the authors compare parameter efficiency with old baselines, why not compare with recent baselines from 2021 such as FEDformer?
>
> **A2**. Since our model is based on graph convolutions, we compare our model with the SOTA GNN-based models. We further conduct an efficiency comparison (under the same input length and prediction length in Table 3) among Transformer-based models on Traffic dataset, the results are as follow:
>
> |Models | parameters| epoch time(s)|Models | parameters| epoch time(s)|
> |:----:|:----:|:----:|:----:|:----:|:----:|
> |FEDformer(2022)| 16211906 | 17.68$\pm$0.49 |AGCRN(2020)|749940|113.46$\pm$1.91|
> |Autoformer(2021)|15425474 | 13.10$\pm$0.71 |StemGNN(2020)|1606140|185.86$\pm$2.22|
> |Informer(2021)|14750658 | 11.76$\pm$0.52 |GraphWaveNet(2019)|280860|105.38$\pm$1.24|
> | | | |Our model |190564|99.25$\pm$1.07|
>
> From the table, we can find that Transformer-based models have larger parameter volumes than GNN-based models while their training time is lower than GNN-based models. This is reasonable and intuitive because 1) it is time-consuming for GNN-based models to construct variable graphs explicitly/implicitly for capturing the spatial dependencies between time series variables in addition to capturing the temporal dependencies; 2) Transformer-based models have a large number of parameters derived from their multi-head self-attention computation and feed-forward networks. In addition, the supra-graph adopted in our EV-FGN is an tremendous graph with $N\*T$ nodes and a graph size of $(N\*T)\times (N\*T)$. However, other GNN-based models construct their graphs over time series variables and are associated with $N$ nodes and a graph size of $N\times N$, and the Transformer-based models corresponds to the time series length, i.e., $T$. In light of the graph size or input size, the efficiency improvement of EV-FGN in terms of training time costs and parameter volumes is more significant and obvious.
>
> **Q3**. Figure 1 can not visualize the proposed method clearly.
>
> **A3**. In Figure 1, we aim to illustrate the overall architecture of our proposed model. As stated in Section 3.1, we 1) first embed MTS inputs to obtain input embeddings; 2) then perform 2D FFT over each discrete $N\times T$ spatial-temporal plane; 3) perform $K$-layer Edge-Varying Fourier Graph Networks (EV-FGN) to achieve multi-layer graph convolutions; 4) perform 2D IFFT on the output of EV-FGN over the according spatial-temporal plane; 5) adopt FFN to generate the multi-step forecasting. In Figure 1, we separate the computation in the Fourier space and the time space. **We will add more descriptions in the caption in the revised version**.
>
> Thank you again for your valuable time. We will carefully polish our manuscript in the revised version according to your comments.

---

### Author Response · Authors · 2022-11-17
**Summary of revision**

Dear Reviewers and Area Chair,

Thanks for your efforts and valuable time on our manuscript.

We address all concerns raised by reviewers and we have updated the new version of the manuscript corresponding to your concerns and questions, where the revised contents are highlighted in blue for ease of reading.

Firstly, we thoroughly proofread the manuscript and corrected the typo in the manuscript, including 'META-LA' (in Table 1) and 'weigh' pointed out by the reviewers. Then, we carefully polished the manuscript according to the comments of the reviewers to make it better qualify this highly selected community.

**Abstract**

We have enhanced the motivation by reorganizing the abstraction (Response to Reviewer VMXz, iyiB).

**Introduction**

First, to provide a better introduction to the motivation, we propose the supra-graph concept after describing the challenges of GNN-based models (Response to Reviewer VMXz, iyiB), and we add an explanation for the "high-resolution" (Response to Reviewer zvHq).

Second, before introducing FGSO, we explain the existing computational problem over the supra-graph in the time domain and why we define an FGSO in the frequency domain. After that, we describe how to conduct multi-layer graph convolutions in the frequency domain. (Response to Reviewer iyiB)

Third, we summarize our contributions. We make the first step to reformulate the graph convolutions and design a complex-valued feed-forward network to efficiently compute multi-layer graph convolutions in the frequency domain (Response to Reviewer VMXz, zvHq, iyiB, jKjm, cY1E).

**Related work**

We have additionally discussed one referred model FEDformer in Section 2.1, we summarize the difference between FEDformer and our model (Response to Reviewer zvHq).

**Methodology**

We have redrawn Fig. 1 and added more captions (Response to Reviewer VMXz, zvHq).

We add an explanation between $n$ and $N\times T$ (Response to Reviewer zvHq).

We correct FGSO to be frequency-invariant. And we add more explanations on the FGSO and the extension to the 2D domain in Appendix C. Many thanks to Reviewer jKjm!

We add more explanations to Eq. 9 (Response to Reviewer jKjm).

**Experiments**

We add a SOTA time series representation learning model, CoST, as our baselines (Response to iyiB). We compare CoST with our model on two datasets (COVID-19 and METR-LA). The results are reported in Appendix G and we further analyze the results.

**Appendix**

We add more explanations about FGSO and experiments in our Appendix.

We add the interpretation of Frequency-invariant FGSO, the explanation of the extension of Definition 2 to the 2D domain, and the interpretation of omitting the weight matrix $W$ in Appendix C.

We add one more baseline, CoST, and report the results in Appendix G.

**Source Code**

We add annotations to explain how to conduct complex-valued multiplications in the model source code (Response to Reviewer zvHq).

We add annotations to describe which data_loader to use for which datasets in the data_loader and main codes (Response to Reviewer jKjm).

Thanks for your valuable advice and suggestions! If you have any questions or concerns, please feel free to let us know.

Best regards.

Authors of Paper 1072.

---

### Decision · Program_Chairs · 2023-01-20

**Decision:**

Reject

**Justification For Why Not Higher Score:**

* Limited novelty, pointed out by several reviewers. Key ideas existed before and the motivation can be considerably improved.
* The high efficiency of FGSO is not obvious, and experimental improvements are often marginal.
* The paper can be difficult to follow. The presentation and clarity of concepts can be improved. This paper is not easy to understand.



**Justification For Why Not Lower Score:**

N/A.

**Metareview: Summary, Strengths And Weaknesses:**

This paper introduces a method for adaptive learning of a supra-graph to represent varying, time-dependent correlations between any two variables at any two time-stamps, to capture high-resolution spatial-temporal dependencies. The supra-graph is learned as a Fourier Graph Shift Operator (GSO) that performs graph convolution in the frequency domain. Corresponding edge-varying diffusion networks are introduced for modeling higher-order diffusion over multiple neighborhood sizes.

While reviewers appreciated that the study is well executed, the agreement is that the idea seems limited, pointing out the following key weaknesses in methodology:
* GSOs and multivariate time series with Fourier transform existed before and the paper extended them to other latent spaces, such as Fourier space.
* The motivation for the method is not convincing enough. The paper does not provide any interpretation for the frequency-invariant parametrization.
* The high efficiency of FGSO is not obvious and experimental improvements are often marginal, though the authors tried to address this issue in the rebuttal phase by performing additional experiments.

An additional point of consensus among reviewers was that reviewers found the paper challenging to follow. The paper is not easy to understand, and the presentation, motivation, and clarity in some concepts can be considerably improved.